# Transforming Gaps into Gains: Bridging Model and Data Heterogeneity in Federated Learning via Knowledge Weak-Aware Zones

**Ke Li**[1], **Yan Ding**[1], **Zhiqin Zhu**[1*], **Shenhai Zheng**[1*]

[1]Chongqing University of Posts and Telecommunications

`{s230301058, s230301006}@stu.cqupt.edu.cn`

`{Zhiqin Zhu, Shenhai Zheng}@cqupt.edu.cn`

## Abstract

Heterogeneous federated learning enables collaborative training across clients under dual heterogeneity of models and data, posing challenges for effective knowledge transfer. Federated mutual learning employs proxy models to bridge cross-model knowledge exchange; however, existing methods remain limited to direct alignment between the outputs of private and proxy models, ignoring the deep discrepancies in representation and decision spaces between them. Such cognitive biases cause knowledge to be transferred only at shallow levels and trigger performance bottlenecks. To address this, this paper proposes FedKWAZ to identify and exploit Knowledge Weak-Aware Zones (KWAZ)—spatial zones of deep knowledge misalignment between private and proxy models, further refined into Semantic Weak-Aware Zones and Decision Weak-Aware Zones, which characterize cognitive misalignments in representation and decision spaces as focal targets for enhanced bidirectional distillation. FedKWAZ designs a Hierarchical Adaptive Patch Mixing (HAPM) mechanism to generate multiple mixed samples and employs a Knowledge Discrepancy Perceptron (KDP) to select the samples exhibiting the largest representation and decision discrepancies, thereby mining critical KWAZ. These modules are integrated into a two-stage mutual learning framework, achieving global class-level representation-decision consistency alignment and local KWAZ-guided refinement, structurally bridging cognitive biases across heterogeneous mutual learning models. Experimental results on multiple datasets and model configurations demonstrate the superior performance of FedKWAZ.

## 1 Introduction

Federated Learning (FL) [21, 9, 25, 22, 24, 38, 37, 39, 19] has emerged as a privacy-preserving distributed training paradigm, enabling devices to collaboratively learn a global model without sharing raw data. However, FL faces two major challenges in practical deployments: clients often hold non-IID data, and necessitate training personalized models that fit device-specific resource constraints. The dual heterogeneity in data and model disrupts traditional parameter aggregation and makes it difficult for a single global model to effectively serve all clients. Consequently, Heterogeneous Federated Learning (HtFL) [44], aiming to learn personalized models for each client, has become a central research focus.

To tackle model and data heterogeneity in HtFL, federated mutual learning [28] has been proposed, where clients train performance-driven private models and resource-friendly proxy models in parallel, facilitating knowledge transfer via bidirectional distillation. However, existing methods [28, 36, 15,

---

[*]Corresponding authors.

26] primarily focus on **direct alignment in the feature and decision space across heterogeneous models, overlooking deeper mismatches in their learned representations and decision behaviors**. As illustrated in Figure 1 (left), for the same input, the private and proxy models on a given client often produce significantly different feature distributions and classification outputs—a phenomenon termed **semantic–decision dual drift**. This dual drift implies that naive output matching yields only coarse and superficial knowledge transfer, leaving critical representation- and decision-level discrepancies unresolved. Without pinpointing and addressing these misaligned "weak zones," knowledge transfer remains inefficient and ultimately constrains the client's performance.

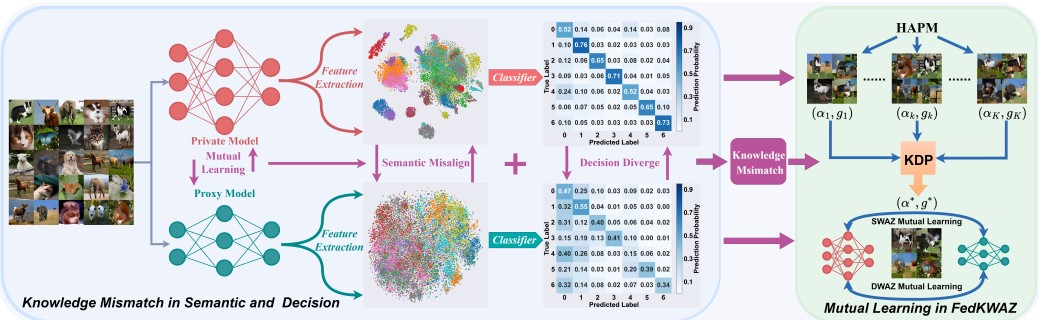

Figure 1: Knowledge mismatch between private and proxy model and mutual learning in FedKWAZ.

Motivated by the necessity to address this semantic–decision dual drift, this paper proposes **Knowledge Weak-Aware Zones (KWAZ)**, characterizing zones where heterogeneous models exhibit the strongest cognitive discrepancies. KWAZ is further decomposed into **Semantic Weak-Aware Zones (SWAZ)** and **Decision Weak-Aware Zones (DWAZ)**, focusing respectively on representation and decision discrepancies that indicate key zones requiring targeted knowledge enhancement. Building on this insight, the FedKWAZ framework is designed to mine and align complementary knowledge within KWAZ through a two-stage mutual learning scheme integrating local and global collaboration.

**At the local stage**, as shown in Figure 1 (right), HAPM generates multiple mixed samples via diverse parameter pairs $(\alpha_i, g_i)$ (where $i = 1, \ldots, K$), where $\alpha$ controls the mixing ratio between two images and $g$ specifies patch granularity to divide each image into $\sqrt{g} \times \sqrt{g}$ blocks for local mixing. KDP then evaluates representation and decision discrepancies between private and proxy models across these samples and selects $(\alpha^*, g^*)$ inducing the maximal divergence, thereby defining the corresponding samples as KWAZ, which are leveraged via SWAZ and DWAZ for path-specific bidirectional distillation, mitigating semantic and decision-level discrepancies between private and proxy models. **At the global stage**, the server aggregates class-level semantic prototypes and decision distributions uploaded by clients to construct unified representation-decision anchors. This approach eliminates the need for frequent proxy model synchronization, significantly reduces communication overhead, and provides representation–decision consistency alignment from a global perspective.

The main contributions of this work are summarized as follows:

- KWAZ is proposed to characterize zones where heterogeneous models exhibit strong cognitive divergence, offering fine-grained perspectives beyond conventional output alignment.
- FedKWAZ is developed as a federated mutual learning framework integrating HAPM and KDP to dynamically discover and exploit SWAZ and DWAZ, enabling complementary learning between private and proxy models in their zones of disagreement.
- A communication-efficient global knowledge alignment mechanism is designed using aggregated class-level semantic and decision anchors, achieving strong cross-client knowledge alignment with significantly reduced communication costs.

## 2  Related Work

HtFL focuses on addressing collaboration challenges arising from variations in clients' computational capabilities, model architectures, and data distributions. Based on the degree of heterogeneity and structural constraints, existing studies can be grouped into the following three categories:

**Resource-Heterogeneous FL Methods.** A number of approaches handle device heterogeneity by allowing clients to train smaller or pruned versions of a global model that fit their local resource constraints. For example, FedRolex [1], FjORD [12], FLASH [4], FedResCuE [47] and HeteroFL [7] enable each client to work with a sub-model or width-reduced network tailored to its hardware capabilities. These methods **focus on resource adaptation** — e.g., dynamically extracting or training sub-networks from a larger global model — so that both high-end and low-end devices can participate in federated training. However, their emphasis is on resource allocation **rather than on maintaining cross-client knowledge consistency**. As a result, resource-heterogeneous solutions do not explicitly align the semantic representations or decision distributions learned by different client models. The absence of such alignment means that knowledge gaps can persist between clients' sub-models, **potentially limiting the overall knowledge transfer efficiency across the federation**.

**Modular-Heterogeneous FL Methods.** Another line of work addresses model heterogeneity by splitting models into shared global modules and private local modules. FedRep [6] and FedPer [3] share and aggregate a global feature extractor across clients (aiming for aligned semantic representations), whereas LG-FedAvg [20] and FedGH [41] share a global classifier across clients (aiming for aligned decision output distributions). By partitioning the model, these approaches attempt to transfer knowledge either at the representation level (feature extractor) or at the decision level (classifier) among clients. However, since the **"global" module in such frameworks is obtained by aggregating components trained on each client's non-iid local data**, forcing all clients to adopt the aggregated module can undermine personalization and lead to suboptimal local performance. In practice, these methods face a **trade-off between achieving global alignment and preserving local adaptation**. Merely sharing or exchanging modules may prove inadequate in addressing fine-grained, data-dependent misalignments in the learned representations or decision boundaries of client models, potentially leaving certain semantic or decision-level discrepancies unresolved.

**Autonomy-heterogeneous methods** allow clients to construct local models without structural limits. Distillation-based approaches, such as FCCL [13] and FedGen [48], leverage public or synthetic data to facilitate cross-model knowledge transfer, yet their effectiveness is highly dependent on data fidelity and distributional consistency. Other methods [30, 31, 44, 45, 14] transmit class prototypes or logits as knowledge carriers. However, since prototypes and logits are information-compressed units, they limit the richness of knowledge transfer and hinder deep, comprehensive consistency alignment.

In contrast, federated mutual learning introduces proxy models as bridges for knowledge transfer, thereby accommodating autonomy-heterogeneous setups. FML [28] and FedKD [36] perform bidirectional distillation between private and proxy models to establish interactive transfer pathways. However, they focus on direct output matching, resulting in shallow distillation that fails to capture critical semantic and decision divergence. FedMRL [42] mitigates this paradigm by incorporating multi-granularity features, yet lacks explicit mechanisms for correcting semantic or structural gaps, introducing noise when concatenating diverse features in its MRL. Moreover, frequent transmissions of proxy models in these methods make it hard to balance communication cost and training gains.

**Insights.** Driven by the limitations identified above—**especially the neglect of deeper semantic and decision discrepancies between heterogeneous models**—this work introduces FedKWAZ, a cognitively-aware mutual learning framework that explicitly targets the problem of semantic–decision dual drift. It formalizes KWAZ as spatial zones where heterogeneous models display significant knowledge divergence, and further decomposes them into SWAZ and DWAZ, enabling targeted distillation in the semantic and decision discrepancy zones, respectively. In addition, a lightweight class-level global knowledge anchoring mechanism is designed to achieve a new balance among architectural flexibility, communication efficiency, and transfer quality. By directly confronting the **semantic–decision dual drift**, FedKWAZ bridges overlooked knowledge gaps and enables more precise and efficient cross-model collaboration.

## 3 The Proposed FedKWAZ Approach

### 3.1 Problem Setting

A typical HtFL system is considered, consisting of a central server and a set of clients $\{\mathcal{C}_0, \mathcal{C}_1, ..., \mathcal{C}_{N-1}\}$. Each client $\mathcal{C}_i$ holds a private local dataset $\mathcal{D}_i$, sampled from a distribution $\mathcal{P}_i$, satisfying $\mathcal{P}_i \neq \mathcal{P}_j$ for any $i \neq j$. In each communication round, $K = N \cdot \rho$ clients are selected according to the participation rate $\rho$. Each client $\mathcal{C}_k$ trains a private model $\mathcal{M}_k$ and a proxy model $\mathcal{Q}_k$

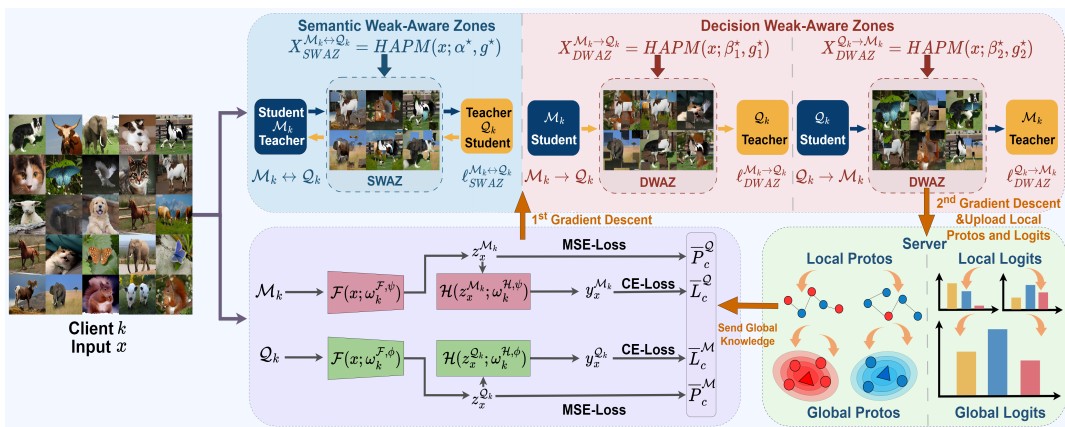

Figure 2: The workflow of FedKWAZ.

on its local data $\mathcal{D}_k$, parameterized by $\psi_k$ and $\phi_k$, respectively. The feature extractor and predictor are defined as $\omega_k^{\mathcal{F}}$ and $\omega_k^{\mathcal{H}}$, represented as $\psi_k = \{\omega_k^{\mathcal{F},\psi}, \omega_k^{\mathcal{H},\psi}\}$ and $\phi_k = \{\omega_k^{\mathcal{F},\phi}, \omega_k^{\mathcal{H},\phi}\}$. Unlike conventional mutual learning, no global model is maintained at the server; instead, global guidance is constructed from feature- and decision-level knowledge uploaded by the clients. FedKWAZ aims to minimize the aggregated local losses across all joint models. The joint model at each client is denoted by $\mathcal{W}_k(\psi_k, \phi_k)$, and the optimization objective is:

$$\min_{\psi_0,\dots,\psi_{N-1},\phi_0,\dots,\phi_{N-1}} \sum_{k=0}^{N-1} \mathbb{E}_{\mathcal{D}_k \sim P_k} \left[ \ell \left( \mathcal{W}_k(\psi_k, \phi_k); \mathcal{D}_k \right) \right] \quad (1)$$

where $\ell(\cdot)$ denotes the empirical loss of the joint model on the client's private dataset. After training, each client retains its private model $\mathcal{M}_k$ for final deployment and inference.

### 3.2 Modeling of KWAZ and Dual-stage Deep Mutual Learning

To investigate cognitive gaps between heterogeneous mutual learning models, KWAZ is characterized as the knowledge-deficient zones of the student model $\mathcal{S}$ and the teacher model $\mathcal{T}$ in both representation and decision spaces. To detect these zones, the following modules are introduced:

- **HAPM**: **Every $k$ rounds**, HAPM generates multiple mixed inputs by partitioning $x$ into $\sqrt{g} \times \sqrt{g}$ patches (granularity $g$) and applying patch-wise interpolation with mixing ratio $\alpha$ under varied $(\alpha, g)$ pairs, enabling discrepancy-aware selection (details in Appendix A).

- **KDP**: For each mixed input generated by HAPM under a specific $(\alpha, g)$, KDP quantifies the knowledge gap between $\mathcal{S}$ and $\mathcal{T}$ on it, and selects $(\alpha^\star, g^\star)$ inducing the largest discrepancy to guide targeted KWAZ mining and refinement **over the following $k$ training rounds**.

Based on this mechanism, a dual-stage mutual learning framework is constructed: coarse-grained cross-client knowledge alignment is achieved at the first stage via global semantic prototypes and decision anchors, while fine-grained distillation is conducted at the second stage under KWAZ guidance. Figure 2 illustrates the workflow, and Appendix B details the algorithm.

#### 3.2.1 Stage I: Global Knowledge Mutual Learning

Global semantic and decision anchors are constructed by aggregating class-level features and predictions from private and proxy models on the server, enabling $\mathcal{M}_k$ and $\mathcal{Q}_k$ to mutually align with each other's global semantic and decision knowledge. Given class-wise dataset $D_{k,c}$ from client $\mathcal{C}_k$, for each $x \in D_{k,c}$, features and predictions from $\mathcal{M}_k$ and $\mathcal{Q}_k$ are:

$$z_x^{\mathcal{M}_k} = \mathcal{F}(x; \omega_k^{\mathcal{F},\psi}), \hat{y}_x^{\mathcal{M}_k} = \mathcal{H}(z_x^{\mathcal{M}_k}; \omega_k^{\mathcal{H},\psi}), z_x^{\mathcal{Q}_k} = \mathcal{F}(x; \omega_k^{\mathcal{F},\phi}), \hat{y}_x^{\mathcal{Q}_k} = \mathcal{H}(z_x^{\mathcal{Q}_k}; \omega_k^{\mathcal{H},\phi}) \quad (2)$$

Class-wise features and logits are averaged as local knowledge. In the case of $\mathcal{M}_k$:

$$P_c^{\mathcal{M}_k} = \frac{1}{|\mathcal{D}_{k,c}|} \sum_{x \in \mathcal{D}_{k,c}} z_x^{\mathcal{M}_k}, \quad L_c^{\mathcal{M}_k} = \frac{1}{|\mathcal{D}_{k,c}|} \sum_{x \in \mathcal{D}_{k,c}} \widehat{y}_x^{\mathcal{M}_k} \tag{3}$$

and the same formulation applies to $\mathcal{Q}_k$, yielding $P_c^{\mathcal{Q}_k}$ and $L_c^{\mathcal{Q}_k}$. All uploaded local knowledge is aggregated to obtain global knowledge, where $\mathcal{N}_c$ denotes clients with data containing class $c$:

$$\bar{P}_c^{\mathcal{M}} = \frac{1}{|\mathcal{N}_c|} \sum_{k \in \mathcal{N}_c} P_c^{\mathcal{M}_k}, \quad \bar{L}_c^{\mathcal{M}} = \frac{1}{|\mathcal{N}_c|} \sum_{k \in \mathcal{N}_c} L_c^{\mathcal{M}_k} \tag{4}$$

with $\bar{P}_c^{\mathcal{Q}}$ and $\bar{L}_c^{\mathcal{Q}}$ aggregated analogously from $\mathcal{Q}_k$. With the global knowledge above, global mutual alignment between $\mathcal{M}_k$ and $\mathcal{Q}_k$ is achieved, where "$\mathcal{M}_k \to \mathcal{Q}_k$" indicates that $\mathcal{M}_k$ learns from $\mathcal{Q}_k$, $\ell_{ce}$ denotes the cross-entropy loss, and $\sigma$ is the softmax function:

$$\ell_k^{\mathcal{M}_k \to \mathcal{Q}_k} = \sum_{c=1}^{C} \sum_{x \in D_{k,c}} \left[ \ell_{ce}(\hat{y}_x^{\mathcal{M}_k}, y) + \ell_{ce}\left(\hat{y}_x^{\mathcal{M}_k}, \sigma(\bar{L}_c^{\mathcal{Q}})\right) + \| z_x^{\mathcal{M}_k} - \bar{P}_c^{\mathcal{Q}} \|_2^2 \right]$$

$$\ell_k^{\mathcal{Q}_k \to \mathcal{M}_k} = \sum_{c=1}^{C} \sum_{x \in D_{k,c}} \left[ \ell_{ce}(\hat{y}_x^{\mathcal{Q}_k}, y) + \ell_{ce}\left(\hat{y}_x^{\mathcal{Q}_k}, \sigma(\bar{L}_c^{\mathcal{M}})\right) + \| z_x^{\mathcal{Q}_k} - \bar{P}_c^{\mathcal{M}} \|_2^2 \right] \tag{5}$$

Model parameters are updated based on the above loss by performing one step of gradient descent:

$$\psi_k^t \leftarrow \psi_k^{t-1} - \eta_\psi \nabla \ell_k^{\mathcal{M}_k \to \mathcal{Q}_k}, \phi_k^t \leftarrow \phi_k^{t-1} - \eta_\phi \nabla \ell_k^{\mathcal{Q}_k \to \mathcal{M}_k} \tag{6}$$

### 3.2.2 Stage II: Local KWAZ Mutual Learning

Following global coarse-grained alignment, FedKWAZ performs local fine-grained mutual learning using KWAZ guidance. This stage incorporates three components: (1) a basic mutual alignment loss, (2) a semantic refinement loss based on SWAZ, and (3) a decision refinement loss based on DWAZ. These objectives are optimized through a second local gradient descent step. Prior to KWAZ-specific distillation, a basic bidirectional loss is computed to align intermediate features and output distributions between $\mathcal{M}_k$ and $\mathcal{Q}_k$. Here, $\text{KL}(p \| q) = \sum_{c=1}^{C} p_c \log \frac{p_c}{q_c}$ denotes the Kullback-Leibler divergence [11], and $\tau$ is a temperature parameter used to smooth the probability outputs:

$$\ell_{\text{base}}^{\mathcal{M}_k \to \mathcal{Q}_k} = \text{KL}\left(\sigma(\widehat{y}_x^{\mathcal{M}_k}/\tau) \| \sigma(\widehat{y}_x^{\mathcal{Q}_k}/\tau)\right) \cdot \tau^2 + \|z_x^{\mathcal{M}_k} - z_x^{\mathcal{Q}_k}\|_2^2$$

$$\ell_{\text{base}}^{\mathcal{Q}_k \to \mathcal{M}_k} = \text{KL}\left(\sigma(\widehat{y}_x^{\mathcal{Q}_k}/\tau) \| \sigma(\widehat{y}_x^{\mathcal{M}_k}/\tau)\right) \cdot \tau^2 + \|z_x^{\mathcal{Q}_k} - z_x^{\mathcal{M}_k}\|_2^2 \tag{7}$$

**SWAZ Mutual Learning: Symmetric Semantic Enhancement.** SWAZ characterizes input zones where student model $\mathcal{S}$ severely mismatches teacher model $\mathcal{T}$ in feature space. To identify such zones, a set of mix ratios $A = \{\alpha_1, \dots, \alpha_m\}$ and patch sizes $G = \{g_1, \dots, g_m\}$ are defined for HAPM. For each $(\alpha, g) \in A \times G$, a mixed input $X_{\text{SWAZ}}^{\mathcal{S} \to \mathcal{T}}$ is generated by blending the original sample $x$ with ratio $\alpha$ and patch size $g$ through HAPM. This mixed sample is fed into both $\mathcal{S}$ and $\mathcal{T}$, producing feature vectors $z_{X_{\text{SWAZ}}^{\mathcal{S} \to \mathcal{T}}}^{\mathcal{S}}$ and $z_{X_{\text{SWAZ}}^{\mathcal{S} \to \mathcal{T}}}^{\mathcal{T}}$. The feature gap between $\mathcal{S}$ and $\mathcal{T}$ on the mixed sample is measured by Mean Square Error (MSE) [33]:

$$\mathcal{L}_{\text{SWAZ}}^{\mathcal{S} \to \mathcal{T}}(\alpha, g; \mathcal{D}) = \mathbb{E}_{x \sim \mathcal{D}} \left[ \|z_{X_{\text{SWAZ}}^{\mathcal{S} \to \mathcal{T}}}^{\mathcal{S}} - z_{X_{\text{SWAZ}}^{\mathcal{S} \to \mathcal{T}}}^{\mathcal{T}}) \|_2^2 \right] \tag{8}$$

The KDP conducts a search over $(\alpha, g) \in A \times G$ to find the pair that maximizes this feature gap:

$$(\alpha^\star, g^\star) = \arg \max_{(\alpha, g) \in A \times G} \mathcal{L}_{\text{SWAZ}}^{\mathcal{S} \to \mathcal{T}}(\alpha, g; \mathcal{D}) \tag{9}$$

Using the optimal pair $(\alpha^\star, g^\star)$, the SWAZ is defined as $X_{\text{SWAZ}}^{\mathcal{M}_k \leftrightarrow \mathcal{Q}_k} = \text{HAPM}(x; \alpha^\star, g^\star)$, Here, "$\mathcal{M}_k \leftrightarrow \mathcal{Q}_k$" indicates that the same input is symmetrically used in both directions, since the optimal parameters are selected by maximizing the MSE-based feature gap between the two models, and MSE is symmetric with respect to its inputs. As a result, exchanging student and teacher roles yields the same mixing parameters and the same SWAZ. Let $z_{X_{\text{SWAZ}}^{\mathcal{M}_k \leftrightarrow Q_k}}^{\mathcal{M}_k}$ and $z_{X_{\text{SWAZ}}^{\mathcal{M}_k \leftrightarrow Q_k}}^{\mathcal{Q}_k}$ denote the feature vectors of $\mathcal{M}_k$ and $\mathcal{Q}_k$ on this optimal mixed input. Then the SWAZ loss for both directions is:

$$\ell_{\text{SWAZ}}^{\mathcal{M}_k \to \mathcal{Q}_k} = \|z_{X_{\text{SWAZ}}^{\mathcal{M}_k \leftrightarrow \mathcal{Q}_k}}^{\mathcal{M}_k} - z_{X_{\text{SWAZ}}^{\mathcal{M}_k \leftrightarrow \mathcal{Q}_k}}^{\mathcal{Q}_k}\|_2^2 = \ell_{\text{SWAZ}}^{\mathcal{Q}_k \to \mathcal{M}_k} = \|z_{X_{\text{SWAZ}}^{\mathcal{M}_k \leftrightarrow \mathcal{Q}_k}}^{\mathcal{Q}_k} - z_{X_{\text{SWAZ}}^{\mathcal{M}_k \leftrightarrow \mathcal{Q}_k}}^{\mathcal{M}_k}\|_2^2 \quad (10)$$

which highlights the symmetric nature of the loss. Minimizing this term encourages $\mathcal{M}_k$ and $\mathcal{Q}_k$ to align their representations in zones where their feature discrepancies are most pronounced.

**DWAZ Mutual Learning: Asymmetric Decision Enhancement.** DWAZ characterizes zones where the predictive distributions of the student model $\mathcal{S}$ and teacher model $\mathcal{T}$ diverge most in the decision space. To expose such zones, a set of mix ratios $B = \{\beta_1, \ldots, \beta_m\}$ and patch sizes $G = \{g_1, \ldots, g_m\}$ are defined for HAPM. For each $(\beta, g) \in B \times G$, a mixed sample $X_{\text{DWAZ}}^{\mathcal{S} \to \mathcal{T}} = \text{HAPM}(x; \beta, g)$ is constructed and passed through $\mathcal{S}$ and $\mathcal{T}$ to obtain logits $\hat{y}_{X_{\text{DWAZ}}^{\mathcal{S} \to \mathcal{T}}}^{\mathcal{S}}$ and $\hat{y}_{X_{\text{DWAZ}}^{\mathcal{S} \to \mathcal{T}}}^{\mathcal{T}}$. The predictive gap is measured using KL divergence:

$$\mathcal{L}_{\text{DWAZ}}^{\mathcal{S} \to \mathcal{T}}(\beta, g; \mathcal{D}) = \mathbb{E}_{x \sim D} \left[ \text{KL} \left( \sigma(\hat{y}_{X_{\text{DWAZ}}^{\mathcal{S} \to \mathcal{T}}}^{\mathcal{S}}/\tau) \| \sigma(\hat{y}_{X_{\text{DWAZ}}^{\mathcal{S} \to \mathcal{T}}}^{\mathcal{T}}/\tau) \right) \cdot \tau^2 \right] \quad (11)$$

The KDP explores $(\beta, g) \in B \times G$ to identify the pair that maximizes the prediction divergence:

$$(\beta^\star, g^\star) = \arg \max_{(\beta, g) \in B \times G} \mathcal{L}_{\text{DWAZ}}^{S \to T}(\beta, g; \mathcal{D}) \quad (12)$$

Unlike SWAZ, DWAZ is direction-dependent due to the asymmetry of KL divergence, where $\text{KL}(p \| q) \neq \text{KL}(q \| p)$. Accordingly, two distinct mixing setups are derived based on the student–teacher role assignment between $\mathcal{M}_k$ and $\mathcal{Q}_k$: $(\beta_1^\star, g_1^\star)$ when $\mathcal{M}_k$ learns from $\mathcal{Q}_k$ (i.e., $\mathcal{S} = \mathcal{M}_k, \mathcal{T} = \mathcal{Q}_k$), and $(\beta_2^\star, g_2^\star)$ when the roles are reversed. These configurations respectively localize the zones where $\mathcal{M}_k$ and $\mathcal{Q}_k$ diverge most from each other's predictions.

When $\mathcal{S} = \mathcal{M}_k$ and $\mathcal{T} = \mathcal{Q}_k$, the corresponding mixed sample is $X_{\text{DWAZ}}^{\mathcal{M}_k \to \mathcal{Q}_k} = \text{HAPM}(x; \beta_1^\star, g_1^\star)$, and the DWAZ loss for the direction $\mathcal{M}_k \to \mathcal{Q}_k$ is:

$$\ell_{\text{DWAZ}}^{\mathcal{M}_k \to \mathcal{Q}_k} = \text{KL} \left( \sigma(\hat{y}_{X_{\text{DWAZ}}^{\mathcal{M}_k \to \mathcal{Q}_k}}^{\mathcal{M}_k}/\tau) \| \sigma(\hat{y}_{X_{\text{DWAZ}}^{\mathcal{M}_k \to \mathcal{Q}_k}}^{\mathcal{Q}_k}/\tau) \right) \cdot \tau^2 \quad (13)$$

Similarly, for the reversed roles where $\mathcal{S} = \mathcal{Q}_k$ and $\mathcal{T} = \mathcal{M}_k$, the mixed input becomes $X_{\text{DWAZ}}^{\mathcal{Q}_k \to \mathcal{M}_k} = \text{HAPM}(x; \beta_2^\star, g_2^\star)$, and the corresponding DWAZ loss for the direction $\mathcal{Q}_k \to \mathcal{M}_k$ is:

$$\ell_{\text{DWAZ}}^{\mathcal{Q}_k \to \mathcal{M}_k} = \text{KL} \left( \sigma(\hat{y}_{X_{\text{DWAZ}}^{\mathcal{Q}_k \to \mathcal{M}_k}}^{\mathcal{Q}_k}/\tau) \| \sigma(\hat{y}_{X_{\text{DWAZ}}^{\mathcal{Q}_k \to \mathcal{M}_k}}^{\mathcal{M}_k}/\tau) \right) \cdot \tau^2 \quad (14)$$

Minimizing $\ell_{\text{DWAZ}}$ in each direction forces $\mathcal{M}_k$ and $\mathcal{Q}_k$ to mutually align their predictions, focusing on zones exhibiting the largest disparity in their decision outputs.

Based on the combination of base mutual loss, SWAZ loss, and DWAZ loss, a second-stage local gradient descent is conducted to refine both representation and decision knowledge between heterogeneous models. The model parameters are updated as follows:

$$\begin{aligned} \psi_k^t &\leftarrow \psi_k^t - \eta_\psi \nabla \left[ \ell_{\text{base}}^{\mathcal{M}_k \to \mathcal{Q}_k} + \ell_{\text{SWAZ}}^{\mathcal{M}_k \to \mathcal{Q}_k} + \ell_{\text{DWAZ}}^{\mathcal{M}_k \to \mathcal{Q}_k} \right] \\ \phi_k^t &\leftarrow \phi_k^t - \eta_\phi \nabla \left[ \ell_{\text{base}}^{\mathcal{Q}_k \to \mathcal{M}_k} + \ell_{\text{SWAZ}}^{\mathcal{Q}_k \to \mathcal{M}_k} + \ell_{\text{DWAZ}}^{\mathcal{Q}_k \to \mathcal{M}_k} \right] \end{aligned} \quad (15)$$

By combining global coarse alignment and local KWAZ-guided refinement, FedKWAZ constructs a two-stage closed-loop distillation process. Heterogeneous models mutually compensate for semantic and decision-level gaps by emphasizing training on KWAZ.

# 4 Experimental Evaluation

## 4.1 Experiment Setup

**Baselines and Datasets.** Ten mainstream heterogeneous federated learning methods are selected for comparison, including LG-FedAvg [20], FedGen [48], FML [28], FedKD [36], FedDistill[14], FedProto [30], FedGH [41], FedMRL [42], FedTGP [44], and FedKTL [45]. Local Training is adopted as the baseline for the non-federated scenario where training is conducted entirely locally. Five public multi-class datasets, covering tasks from natural image to medical image, are used. The datasets are Cifar10, Cifar100 [17], Skin-Lesions-14 (merged from HAM10000 [32] and MSLDv2.0 [2]), Flowers102 [23], and Tiny-ImageNet [5].

**Training Configuration.** The experimental setup involves 20 clients, all participating in each communication round ($\rho = 1$). Local training is performed once per round using a learning rate of 0.01 and a batch size of 10, over a total of 1000 rounds. Each client splits its local dataset into 75% for training and 25% for testing. The reported result corresponds to the optimal averaged test accuracy across all clients at each round. All experiments are repeated three times, and both the mean and standard deviation are reported.

**Data Heterogeneity Modeling.** Follow FedALA [43], two non-IID partitioning schemes are used to simulate data heterogeneity: **Pathological Split:** For Cifar10, Skin-Lesions-14, Cifar100, Flowers102, and Tiny-ImageNet (with 10/14/100/102/200 classes, respectively), each client receives 2/2/10/10/20 disjoint classes to induce class-wise isolation. **Practical Split:** For each class $c$ and client $i$, the sample proportion is drawn from a Dirichlet distribution $q_{c,i} \sim Dir(\beta)$, with $\beta = 0.1$, introducing heterogeneity in both class distribution and data volume.

**Modeling heterogeneity scenarios.** Two types of heterogeneous setups are constructed for feature extractors and classifiers. **Feature extractors** follow the HtFE$_X$ scheme, where client $i$ is assigned the $(i \bmod X)$-th model. An average pooling layer with output dimension $K = 512$ is applied for feature alignment. The main test setup, HtFE$_8$, including 4-layer CNN [21], GoogleNet [29], MobileNet_v2 [27], and ResNet variants [10]. **For classifiers**, together with HtFE$_8$, two Transformer architectures—ViT-B/16 and ViT-B/32 [8]—with different classifier architectures of ResNets, form the HtM$_{10}$ setup. Furthermore, the HtFE$_8$–HtC$_4$ introduces dual heterogeneity by pairing the eight extractors with four distinct fully connected classifiers. Methods requiring homogeneous classifier structures (LG-FedAvg, FedGen, FedGH) are excluded from this comparison. All architectural details and extended results are provided in Appendix D and Appendix E.

## 4.2 Performance under Two Data Heterogeneity Settings

As shown in Table 1, FedKWAZ yields accuracy improvements of $4.54\%$ and $3.80\%$ on Cifar100 and Flowers102 under the Dirichlet split. For the Pathological scenario, further gains of $4.66\%$ and $3.87\%$ are reported on Cifar100 and Tiny-ImageNet. These benefits arise from KWAZ-driven knowledge refinement, by adaptively targeting and supervising cognitively misaligned samples via SWAZ and DWAZ, FedKWAZ effectively addresses heterogeneous label assignment and sample imbalance. Performance trends in Figure 3 confirm its stability across tasks.

Table 1: Accuracy across datasets with top three results highlighted as **first** , second , and third .

| Settings | Practical Setting | | | | | Pathological Setting | | | | |
|---|---|---|---|---|---|---|---|---|---|---|
| Datasets | Cifar10 | Skin-Lesions-14 | Cifar100 | Flowers102 | Tiny-ImageNet | Cifar10 | Skin-Lesions-14 | Cifar100 | Flowers102 | Tiny-ImageNet |
| Local | 86.14±0.18 | 86.73±0.27 | 39.50±0.21 | 49.00±0.14 | 23.53±0.07 | 84.15±0.13 | 93.63±0.18 | 56.27±0.50 | 59.56±0.36 | 31.87±0.06 |
| FedDistill | 86.49±0.11 | 86.88±0.20 | 41.58±0.37 | 49.05±0.17 | 25.21±0.16 | 84.59±0.35 | 93.02±0.23 | 57.89±0.33 | 63.84±0.40 | 33.10±0.08 |
| LG-FedAvg | 85.03±0.11 | 85.40±0.18 | 41.13±0.22 | 46.33±0.22 | 23.40±0.35 | 84.50±0.17 | 93.15±0.31 | 55.58±0.13 | 58.52±0.34 | 32.50±0.15 |
| FedGen | 85.22±0.12 | 86.47±0.19 | 40.09±0.20 | 49.39±0.24 | 23.69±0.19 | 84.39±0.26 | 93.01±0.18 | 55.53±0.13 | 59.16±0.29 | 32.03±0.10 |
| FedKD | 86.31±0.11 | 86.66±0.20 | 40.37±0.18 | 46.67±0.15 | 24.73±0.08 | 84.62±0.12 | 93.25±0.18 | 54.55±0.10 | 59.66±0.28 | 32.40±0.06 |
| FedProto | 84.57±0.13 | 81.57±0.26 | 35.83±0.15 | 41.08±0.14 | 18.37±0.32 | 83.49±0.21 | 92.28±0.19 | 52.66±0.11 | 56.55±0.30 | 28.41±0.10 |
| FML | 86.44±0.10 | 86.84±0.18 | 39.47±0.20 | 45.35±0.17 | 23.56±0.18 | 83.60±0.12 | 93.12±0.31 | 53.26±0.15 | 58.28±0.28 | 31.60±0.14 |
| FedGH | 83.77±0.48 | 83.99±0.25 | 38.83±0.11 | 45.94±0.15 | 22.60±0.12 | 83.39±0.15 | 93.29±0.19 | 56.25±0.13 | 59.41±0.36 | 31.77±0.08 |
| FedMRL | 86.79±0.18 | 87.60±0.25 | 42.22±0.27 | 48.47±0.16 | 23.37±0.22 | 85.80±0.21 | 94.33±0.24 | 58.55±0.23 | 62.41±0.30 | 33.48±0.17 |
| FedTGP | 88.07±0.42 | 87.33±0.36 | 47.15±0.17 | 53.72±0.23 | 27.62±0.08 | 87.03±0.13 | 94.66±0.18 | 61.73±0.32 | 69.07±0.41 | 34.70±0.08 |
| FedKTL | 88.02±0.13 | 87.23±0.24 | 47.09±0.16 | 52.85±0.29 | 28.17±0.22 | 86.21±0.34 | 93.70±0.19 | 61.20±0.16 | 64.19±0.45 | 34.74±0.32 |
| FedKWAZ | **90.39±0.12** | **89.82±0.19** | **51.69±0.14** | **57.52±0.21** | **30.85±0.09** | **90.17±0.11** | **96.27±0.08** | **66.39±0.17** | **71.28±0.29** | **38.61±0.12** |

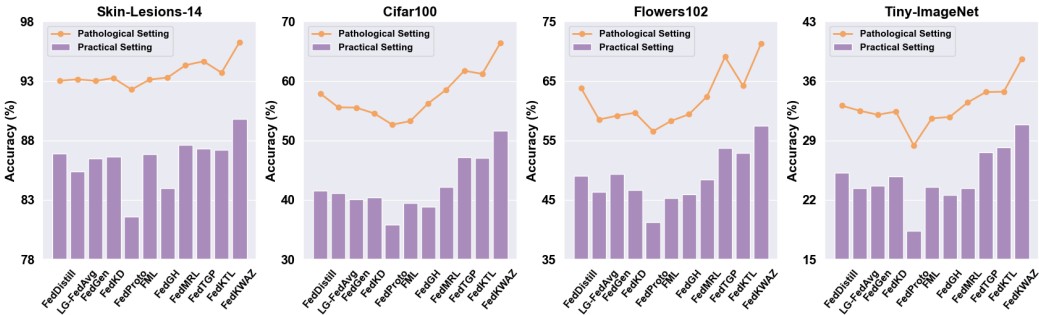

Figure 3: Performance comparison under pathological and dirichlet data heterogeneity settings.

## 4.3 Performance under Various Model Heterogeneity Settings

As shown in Table 2, most methods experience noticeable performance degradation as model heterogeneity increases, particularly during the transition from the homogeneous $HtFE_1$ to the heterogeneous $HtFE_2$. Figure 4 (left one) highlights the sensitivity of cross-client collaboration to architectural variation. In contrast, FedKWAZ performs architecture-aware distillation by decoupling semantic and decision discrepancies, enabling path-specific alignment across heterogeneous models. It yields a $2.45\%$ average gain and outperforms FedTGP by $3.26\%$ under the $HtFE_8$–$HtC_4$ setting, where both extractors and classifiers are heterogeneous, demonstrating high adaptability to model diversity.

Table 2: Effects of model heterogeneity and client scale on Cifar100 test performance.

| Settings | Different Degrees of Model Heterogeneity | | | | | | Large Client Amount | | |
|---|---|---|---|---|---|---|---|---|---|
| | $HtFE_1$ | $HtFE_2$ | $HtFE_4$ | $HtFE_9$ | $HtM_{10}$ | $HtFE_8$-$HtC_4$ | 50 Clients | 100 Clients | 200 Clients |
| Local | 53.64±0.15 | 44.94±0.48 | 43.58±0.21 | 43.46±0.16 | 40.23±0.09 | 39.96±0.23 | 38.28±0.12 | 36.76±0.16 | 31.59±0.12 |
| FedDistill | 53.67±0.12 | 46.38±0.26 | 44.54±0.17 | 44.31±0.27 | 40.45±0.23 | 40.66±0.10 | 40.64±0.38 | 39.69±0.11 | 32.61±0.15 |
| LG-FedAvg | 53.89±0.13 | 45.49±0.14 | 43.74±0.29 | 43.65±0.10 | — | — | 38.54±0.10 | 36.26±0.15 | 30.34±0.11 |
| FedGen | 53.19±0.16 | 44.70±0.17 | 43.26±0.18 | 42.46±0.14 | — | — | 38.42±0.14 | 35.44±0.10 | 30.85±0.14 |
| FedKD | 53.09±0.10 | 46.17±0.12 | 44.60±0.13 | 42.30±0.15 | 40.19±0.11 | 40.30±0.10 | 39.30±0.12 | 34.39±0.33 | 30.34±0.16 |
| FedProto | 44.33±0.45 | 46.10±0.13 | 39.47±0.14 | 31.34±0.12 | 34.23±0.19 | 30.80±0.14 | 34.82±0.25 | 32.24±0.19 | 26.22±0.20 |
| FML | 53.03±0.11 | 43.01±0.16 | 42.79±0.19 | 42.36±0.10 | 40.17±0.11 | 40.43±0.16 | 38.58±0.10 | 36.10±0.11 | 30.75±0.10 |
| FedGH | 49.13±0.23 | 44.52±0.19 | 43.35±0.14 | 41.73±0.20 | — | — | 38.14±0.15 | 33.73±0.14 | 29.27±0.30 |
| FedMRL | 45.08±0.35 | 44.34±0.14 | 43.81±0.25 | 42.85±0.22 | 41.34±0.12 | 41.80±0.11 | 39.24±0.12 | 36.73±0.16 | 31.77±0.15 |
| FedTGP | 50.48±0.24 | 49.91±0.55 | 46.75±0.15 | 48.15±0.30 | 44.02±0.20 | 44.65±0.33 | 43.21±0.25 | 40.90±0.13 | 33.87±0.19 |
| FedKTL | 53.53±0.15 | 47.02±0.14 | 48.00±0.17 | 49.99±0.16 | 45.64±0.16 | 44.42±0.16 | 43.19±0.16 | 39.85±0.15 | 34.29±0.14 |
| FedKWAZ | **54.33±0.17** | **52.05±0.12** | **51.69±0.11** | **52.42±0.23** | **48.35±0.21** | **47.91±0.20** | **47.79±0.12** | **43.70±0.13** | **38.18±0.12** |

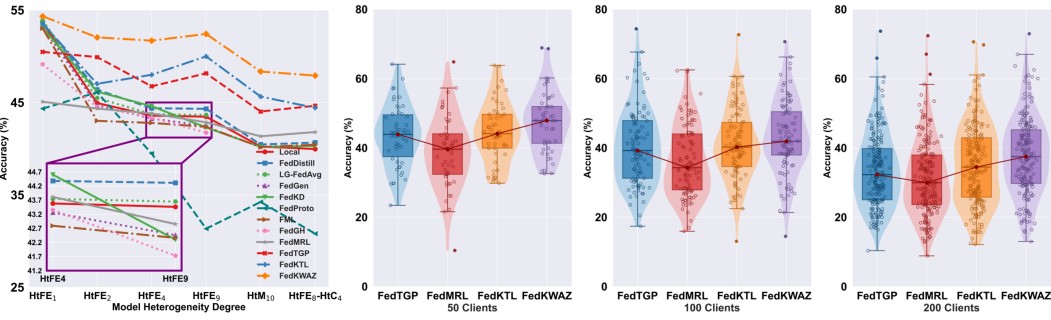

Figure 4: Test accuracy across various model settings (left one) and client scales (right three).

## 4.4 Performance in Multi-Client Setup

Scalability is evaluated with 50, 100, and 200 clients under partial participation ($\rho = 0.5$) to simulate intermittent communication. As shown in Table 4, most methods exhibit notable performance degradation as client count increases and data becomes more fragmented. In contrast, FedKWAZ maintains margins of $4.58\%$, $2.80\%$, and $3.89\%$ across the three scales, benefiting from its capacity to extract transferable cues from HAPM-generated inputs and enhance cognitively divergent zones, thus alleviating data sparsity-induced degradation. The per-client accuracy distributions in Figure 4

(right three) further support this trend, with FedKWAZ presenting higher medians, smaller variances, and more concentrated distribution, indicating improved utility under large-scale deployment.

## 4.5 Communication and Computation Cost

Table 3 reports the theoretical and empirical communication cost per round. In FedGen, the auxiliary generator is denoted as $\Theta$; in FML, FedKD and FedMRL, $\theta_g$ and $w_g$ represent the proxy feature extractor and classifier, with singular value decomposition (SVD) applied to $\theta_g$ at a compression ratio $r$, where $|\theta_g| \gg K \times C$ and $C_i$ is the class count for client $i$. In FedKTL, the generator latent size is $K_G$, and the generated image has $C_h$ channels and spatial size $S$. "M" denotes "million".

Table 3: Communication cost per iteration of HtFE$_8$ model group in theory and practice for Cifar100.

| | Theory | | Practice | | Total |
|---|---|---|---|---|---|
| | Upload | Download | Upload | Download | |
| FedDistill | $\sum_{i=1}^M C \times C_i$ | $M \times C \times C$ | 0.09M | 0.20M | 0.29M |
| LG-FedAvg | $\sum_{i=1}^M |w_i|$ | $\sum_{i=1}^M |w_i|$ | 1.03M | 1.03M | 2.06M |
| FedGen | $\sum_{i=1}^M |w_i|$ | $\sum_{i=1}^M (|w_i| + |\Theta|)$ | 1.03M | 7.66M | 8.69M |
| FedKD | $M \times (|\theta_g| + |w_g|) \times r$ | $M \times (|\theta_g| + |w_g|) \times r$ | 16.52M | 16.52M | 33.04M |
| FedProto | $\sum_{i=1}^M K \times C_i$ | $M \times K \times C$ | 0.44M | 1.02M | 1.46M |
| FML | $M \times (|\theta_g| + |w_g|)$ | $M \times (|\theta_g| + |w_g|)$ | 18.50M | 18.50M | 37.00M |
| FedGH | $\sum_{i=1}^M K \times C_i$ | $\sum_{i=1}^M |w_i|$ | 0.44M | 1.03M | 1.47M |
| FedMRL | $M \times (|\theta_g| + |w_g|)$ | $M \times (|\theta_g| + |w_g|)$ | 18.50M | 18.50M | 37.00M |
| FedTGP | $\sum_{i=1}^M K \times C_i$ | $M \times K \times C$ | 0.44M | 1.02M | 1.46M |
| FedKTL | $\sum_{i=1}^M C \times C_i$ | $M \times C \times (K_G + C_h \times S)$ | 0.09M | 7.17M | 7.26M |
| FedKWAZ | $\sum_{i=1}^M (K + C) \times C_i * 2$ | $M \times (K + C) \times C * 2$ | 1.06M | 2.44M | 3.50M |

Substantial transmission overhead is introduced by FML, FedKD, and FedMRL due to the need for proxy model exchange. FedGen further amplifies downstream bandwidth via generator transmission. Although FedDistill maintains low communication volume, its accuracy degrades due to limited information content in transmitted logits. FedKTL also exhibits increased downstream load during generator-based knowledge transfer.

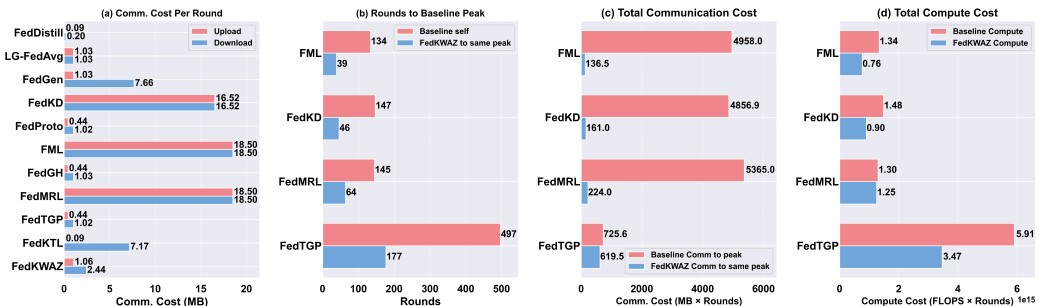

Figure 5: Comparison of communication and computation costs across methods.

As shown in Figure 5 (a), owing to the use of class-level semantic representations and prediction outputs as compact cross-model carriers, FedKWAZ achieves substantial reductions in both upload and download communication volume. Figure 5 (b) reports the number of communication rounds required by FedTGP, FML, FedKD, and FedMRL to reach peak performance, compared to the rounds needed by FedKWAZ to match these baselines. Total FLOPs per local forward-backward pass are also evaluated. Benefiting from its two-stage mutual learning scheme, FedKWAZ attains and surpasses baseline performance with fewer rounds. As illustrated in Figures 5 (c)(d), both communication and computation costs are reduced, improving adaptability under high-frequency interactions.

## 4.6 Ablation Study

Three ablation variants are constructed to isolate the contributions of FedKWAZ components: (1) Local Mutual Learning (LML), which omits the global alignment stage; (2) One-way KWAZ learning from proxy model to private model (Proxy-to-Private Learning, P2P-L); (3) One-way KWAZ learning in reverse from private model to proxy model (Private-to-Proxy Reverse Learning, P2P-R). As reported in Table 4, Fed-

Table 4: Ablation study on four datasets in the practical setting using HtFE$_8$.

| | LML | P2P-L | P2P-R | FedKWAZ |
|---|---|---|---|---|
| Cifar10 | 89.42±0.10 | 89.63±0.15 | 89.95±0.13 | **90.39±0.12** |
| Skin-Lesions-14 | 88.70±0.18 | 89.12±0.22 | 89.30±0.16 | **89.82±0.19** |
| Cifar100 | 50.36±0.16 | 50.93±0.12 | 51.13±0.19 | **51.69±0.14** |
| Flowers102 | 56.44±0.25 | 56.87±0.17 | 57.08±0.23 | **57.52±0.21** |

KWAZ consistently outperforms LML by 0.97%, 1.12%, 1.33%, and 1.08% on Cifar10, Skin-Lesions-14, Cifar100, and Flowers102, respectively. Figure 6 (left) shows that one-way transfer pro-

vides limited gains over LML, suggesting that cross-model knowledge gaps remain under-addressed. In contrast, the full FedKWAZ—with bidirectional KWAZ learning and global alignment, achieves markedly superior results, demonstrating its efficacy in focused and deeper knowledge transfer.

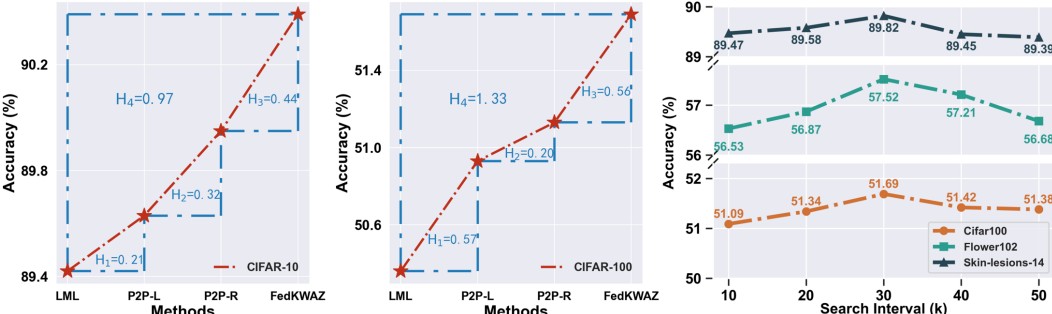

Figure 6: Ablation study results (left two) and hyperparameter study results (right one).

## 4.7 Hyperparameter Study

As shown in Figure 6 (right one), the impact of updating HAPM mixing configurations via KDP every $k$ rounds is evaluated with $k \in \{10, 20, 30, 40, 50\}$. Frequent updates ($k = 10$ or $20$) lead to performance degradation, as the model fails to fully exploit the knowledge learned from KWAZ. Conversely, infrequent updates ($k = 40$ or $50$) hinder the timely detection and bridging of emerging knowledge gaps, limiting the effectiveness of KWAZ-guided learning. Accuracy consistently peaks at $k = 30$ across datasets, suggesting that a balanced schedule between knowledge absorption and dynamic gap correction is essential. Overall, FedKWAZ sustains strong performance across different $k$, validating the effectiveness of its dynamic update strategy.

## 5    Conclusions

This paper proposes FedKWAZ, a cognitively structured mutual learning framework that captures semantic and decision discrepancies across heterogeneous models by decomposing KWAZ into SWAZ and DWAZ. Through HAPM-based sample structuring and KDP-guided discrepancy localization, KWAZ is dynamically identified and reinforced, offering a fine-grained perspective beyond conventional output alignment. Comprehensive experiments across diverse datasets and architectures validate its effectiveness under both model and data heterogeneity in federated learning.

## Acknowledgments and Disclosure of Funding

This work was supported in part by the National Natural Science Foundation of China under Grants 62576062, 62331008 and 62276039, in part by the Science and Technology Research Program of Chongqing Municipal Education Commission under Grants KJZD-M202500604 and KJZD-K202200606, in part by the New Chongqing Youth Innovation Talent under Grant CSTB2024NSCQQCXMX0067.

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

# Appendices

An overview of the Appendix is provided below, covering algorithmic components, theoretical foundations, implementation specifics, extended empirical analysis, and broader contextual considerations.

# A  Hierarchical Adaptive Patch Mixup

To explicitly capture knowledge discrepancies between heterogeneous models, a novel Hierarchical Adaptive Patch Mixup HAPM mechanism is introduced in FedKWAZ. High-quality samples with local diversity and hierarchical structure are generated through spatially granular partitioning and stochastic perturbation, supporting the identification and enhancement of weak knowledge-aware zones. The input to HAPM includes the images $x$, a mixing strength parameter $\alpha$, and a spatial granularity parameter $G$. Two key procedures are involved in HAPM: hierarchical random partitioning of spatial patches and independent interpolation of partitioned samples.

## A.1  Hierarchical Random Partitioning of Spatial Patches

To reflect knowledge differences across heterogeneous models at a local granularity, a spatially hierarchical random perturbation mechanism is incorporated into HAPM. The original input images $X \in \mathbb{R}^{B \times C \times H \times W}$ (where $B$, $C$, $H$, and $W$ denote the batch size, channels, height, and width) is partitioned into $\sqrt{G} \times \sqrt{G}$ local patches. Each patch is then subjected to spatial perturbations to improve diversity and sensitivity to knowledge. Specifically, the base height and width of each patch are defined as:

$$h_{\text{base}} = \left\lfloor \frac{H}{\sqrt{G}} \right\rfloor, \quad w_{\text{base}} = \left\lfloor \frac{W}{\sqrt{G}} \right\rfloor \tag{16}$$

Next, the initial boundaries of each patch are constructed based on grid indices:

$$h_{\text{start}}^{(i,j)} = i \cdot h_{\text{base}}, \quad h_{\text{end}}^{(i,j)} = (i+1) \cdot h_{\text{base}}, \quad w_{\text{start}}^{(i,j)} = j \cdot w_{\text{base}}, \quad w_{\text{end}}^{(i,j)} = (j+1) \cdot w_{\text{base}} \tag{17}$$

Here, the grid indices $i, j = 0, 1, \ldots, \sqrt{G} - 1$ are used to specify patch positions. To increase the diversity of local perturbations, jitter factors $\delta_h^{(i,j)}$ and $\delta_w^{(i,j)}$ are introduced and sampled as integer variables from uniform distributions:

$$\delta_h^{(i,j)} \sim \mathcal{U}\left(-\frac{h_{\text{base}}}{4}, \frac{h_{\text{base}}}{4}\right), \quad \delta_w^{(i,j)} \sim \mathcal{U}\left(-\frac{w_{\text{base}}}{4}, \frac{w_{\text{base}}}{4}\right) \tag{18}$$

To ensure validity, the perturbed positions are clipped within image boundaries using the clip function:

$$\hat{h}_{\text{start}}^{(i,j)} = \text{clip}\left(h_{\text{start}}^{(i,j)} + \delta_h^{(i,j)}, 0, H\right), \hat{h}_{\text{end}}^{(i,j)} = \text{clip}\left(h_{\text{end}}^{(i,j)} + \delta_h^{(i,j)}, 0, H\right)$$
$$\hat{w}_{\text{start}}^{(i,j)} = \text{clip}(w_{\text{start}}^{(i,j)} + \delta_w^{(i,j)}, 0, W), \hat{w}_{\text{end}}^{(i,j)} = \text{clip}(w_{\text{end}}^{(i,j)} + \delta_w^{(i,j)}, 0, W) \tag{19}$$

These designs jointly form the hierarchical structure of HAPM, where local-granularity patches are first divided in space and then perturbed at the patch level to enhance the diversity of the mixed samples.

## A.2  Patch-wise Independent Interpolation and Fusion

Based on the above spatial partitioning, the mixing coefficient $\lambda^{(i,j)}$ is independently sampled for each patch zone from a Beta distribution:

$$\lambda^{(i,j)} \sim \text{Beta}(\alpha, \alpha), \quad \alpha > 0 \tag{20}$$

Given a randomly shuffled batch index $\pi$, the interpolation fusion is independently performed in each patch zone:

$$X^{\text{mix}}[:,:,\hat{h}_{\text{start}}^{(i,j)} : \hat{h}_{\text{end}}^{(i,j)}, \hat{w}_{\text{start}}^{(i,j)} : \hat{w}_{\text{end}}^{(i,j)}] = \lambda^{(i,j)} X[:,:,\hat{h}_{\text{start}}^{(i,j)} : \hat{h}_{\text{end}}^{(i,j)}, \hat{w}_{\text{start}}^{(i,j)} : \hat{w}_{\text{end}}^{(i,j)}]$$
$$+ (1 - \lambda^{(i,j)}) X[\pi,:,\hat{h}_{\text{start}}^{(i,j)} \cdot \hat{h}_{\text{end}}^{(i,j)}, \hat{w}_{\text{start}}^{(i,j)} \cdot \hat{w}_{\text{end}}^{(i,j)}] \tag{21}$$

Through this mixing process, each local zone of the image is independently adjusted based on the patch-specific mixing strength, enabling HAPM to generate more diverse and knowledge-sensitive samples, thereby enhancing the model's recognition capacity in knowledge-weak zones.

# B   Pseudo Codes of FedKWAZ

---

**Algorithm 1:** FedKWAZ

---

**Input:** $N$, total number of clients; $\rho$, participation rate of clients in one round; $T$, total number of rounds; $\eta$, learning rate of private and proxy models; $E_A$,$E_B$, training epoch of the first and second stage; $\alpha^*$, $\beta_1^*$, $\beta_2^*$, $g^*$, mixing parameters of HAPM; $\tau$, temperature of distillation; $\mathcal{D}_k$, the local dataset of the k-th client.
**Output:** Private and proxy models' $P_c^{\mathcal{M}_k}$, $P_c^{\mathcal{Q}_k}$ and $L_c^{\mathcal{M}_k}$, $L_c^{\mathcal{Q}_k}$.
Randomly initialize the client local heterogeneous models $[\mathcal{M}_1(\psi_1), \ldots, \mathcal{M}_{N-1}(\psi_{N-1})]$ and local proxy homogeneous small models $[\mathcal{Q}_1(\phi_1), \ldots, \mathcal{Q}_{N-1}(\phi_{N-1})]$.

**for** *each round t=1,...,T-1* **do**

    **// Server Side**:
    $S^t \leftarrow$ Randomly sample $K$ clients from $N$ clients; Broadcast $\bar{P}_c^{\mathcal{M}}$, $\bar{P}_c^{\mathcal{Q}}$, $\bar{L}_c^{\mathcal{M}}$, $\bar{L}_c^{\mathcal{Q}}$ to them;
    **for** *each client $k = 1, \ldots, K$* **do**
       | ClientUpdate($\bar{P}_c^{\mathcal{M}}$, $\bar{P}_c^{\mathcal{Q}}$, $\bar{L}_c^{\mathcal{M}}$, $\bar{L}_c^{\mathcal{Q}}$)
    **end**
    Aggregate global semantic representation and decision output.
    **// ClientUpdate**:
    Receive $\bar{P}_c^{\mathcal{M}}$, $\bar{P}_c^{\mathcal{Q}}$, $\bar{L}_c^{\mathcal{M}}$, $\bar{L}_c^{\mathcal{Q}}$ from the server;
    **for** $k \in S^t$ **do**

        **// Stage I: Global Knowledge Mutual Learning**
        **for** *epoch $e = 1, \ldots, E_A$* **do**
            **for** *batch $(x, y) \in \mathcal{D}_k$* **do**
                Obtain representation and logits:
                $z_x^{\mathcal{M}_k}, \hat{y}_x^{\mathcal{M}_k} \leftarrow \mathcal{M}_k(x; \psi_k)$, $z_x^{\mathcal{Q}_k}, \hat{y}_x^{\mathcal{Q}_k} \leftarrow \mathcal{Q}_k(x; \phi_k)$;
                Compute global alignment loss:
                $\ell_k^{\mathcal{M}_k \to \mathcal{Q}_k} = \ell_{CE}(\hat{\mathbf{y}}_x^{\mathcal{M}_k}, y) + \ell_{CE}\left(\hat{\mathbf{y}}_x^{\mathcal{M}_k}, \sigma(\bar{L}_c^{\mathcal{Q}})\right) + \| z_x^{\mathcal{M}_k} - \bar{P}_c^{\mathcal{Q}} \|_2^2$;
                $\ell_k^{\mathcal{Q}_k \to \mathcal{M}_k} = \ell_{CE}(\hat{\mathbf{y}}_x^{\mathcal{Q}_k}, y) + \ell_{CE}\left(\hat{\mathbf{y}}_x^{\mathcal{Q}_k}, \sigma(\bar{L}_c^{\mathcal{M}})\right) + \| z_x^{\mathcal{Q}_k} - \bar{P}_c^{\mathcal{M}} \|_2^2$;
                Update: $\psi_k^t \leftarrow \psi_k^{t-1} - \eta_\psi \nabla \ell_k^{\mathcal{M}_k \to \mathcal{Q}_k}$, $\phi_k^t \leftarrow \phi_k^{t-1} - \eta_\phi \nabla \ell_k^{\mathcal{Q}_k \to \mathcal{M}_k}$;
            **end**
        **end**

        **// Stage II: Local KWAZ Mutual Learning**
        **for** *epoch $e = 1, \ldots, E_B$* **do**
            **for** *batch $(x, y) \in \mathcal{D}_k$* **do**
                Compute base mutual learning losses:
                $\ell_{\text{base}}^{\mathcal{M}_k \to \mathcal{Q}_k} = \text{KL}\left(\sigma(\hat{\mathbf{y}}_x^{\mathcal{M}_k}/\tau) \,\|\, \sigma(\hat{\mathbf{y}}_x^{\mathcal{Q}_k}/\tau)\right) \cdot \tau^2 + \|z_x^{\mathcal{M}_k} - z_x^{\mathcal{Q}_k}\|_2^2$;
                $\ell_{\text{base}}^{\mathcal{Q}_k \to \mathcal{M}_k} = \text{KL}\left(\sigma(\hat{\mathbf{y}}_x^{\mathcal{Q}_k}/\tau) \,\|\, \sigma(\hat{\mathbf{y}}_x^{\mathcal{M}_k}/\tau)\right) \cdot \tau^2 + \|z_x^{\mathcal{Q}_k} - z_x^{\mathcal{M}_k}\|_2^2$;
                Generate SWAZ mixed samples: $X_{\text{SWAZ}}^{\mathcal{M}_k \leftrightarrow \mathcal{Q}_k} = \text{HAPM}(x; \alpha^\star, g^\star)$;
                Compute semantic weak-awareness loss:
                $\ell_{\text{SWAZ}}^{\mathcal{M}_k \to \mathcal{Q}_k} = \|z_{X_{\text{SWAZ}}^{\mathcal{M}_k \leftrightarrow \mathcal{Q}_k}}^{\mathcal{M}_k} - z_{X_{\text{SWAZ}}^{\mathcal{M}_k \leftrightarrow \mathcal{Q}_k}}^{\mathcal{Q}_k}\|_2^2 = \ell_{\text{SWAZ}}^{\mathcal{Q}_k \to \mathcal{M}_k}$;
                Generate DWAZ mixed samples:
                $X_{\text{DWAZ}}^{\mathcal{M}_k \to \mathcal{Q}_k} = \text{HAPM}(x; \beta_1^\star, g_1^\star)$, $X_{\text{DWAZ}}^{\mathcal{Q}_k \to \mathcal{M}_k} = \text{HAPM}(x; \beta_2^\star, g_2^\star)$;
                Compute decision weak-awareness losses:
                $\ell_{\text{DWAZ}}^{\mathcal{M}_k \to \mathcal{Q}_k} = \text{KL}\left(\sigma(\hat{y}_{X_{\text{DWAZ}}^{\mathcal{M}_k \to \mathcal{Q}_k}}^{\mathcal{M}_k}/\tau) \,\|\, \sigma(\hat{y}_{X_{\text{DWAZ}}^{\mathcal{M}_k \to \mathcal{Q}_k}}^{\mathcal{Q}_k}/\tau)\right) \cdot \tau^2$;
                $\ell_{\text{DWAZ}}^{\mathcal{Q}_k \to \mathcal{M}_k} = \text{KL}\left(\sigma(\hat{y}_{X_{\text{DWAZ}}^{\mathcal{Q}_k \to \mathcal{M}_k}}^{\mathcal{Q}_k}/\tau) \,\|\, \sigma(\hat{y}_{X_{\text{DWAZ}}^{\mathcal{Q}_k \to \mathcal{M}_k}}^{\mathcal{M}_k}/\tau)\right) \cdot \tau^2$;
                Update models again:
                $\psi_k^t \leftarrow \boldsymbol{\psi}_k^t - \eta_\psi \nabla[\ell_{\text{base}}^{\mathcal{M}_k \to \mathcal{Q}_k} + \ell_{\text{SWAZ}}^{\mathcal{M}_k \to \mathcal{Q}_k} + \ell_{\text{DWAZ}}^{\mathcal{M}_k \to \mathcal{Q}_k}]$;
                $\phi_k^t \leftarrow \phi_k^t - \eta_\phi \nabla[\ell_{\text{base}}^{\mathcal{Q}_k \to \mathcal{M}_k} + \ell_{\text{SWAZ}}^{\mathcal{Q}_k \to \mathcal{M}_k} + \ell_{\text{DWAZ}}^{\mathcal{Q}_k \to \mathcal{M}_k}]$;
            **end**
        **end**

        Aggregate and upload updated $P_c^{\mathcal{M}_k}$, $P_c^{\mathcal{Q}_k}$,$L_c^{\mathcal{M}_k}$, $L_c^{\mathcal{Q}_k}$ to the server.
    **end**
**end**

---

## C Theoretical Proofs

The following notations and expressions are defined. $t \in 0, 1, ..., T-1$ is used to denote the $t$-th round of federated communication. The local loss function $\mathcal{L}_k(w_k)$ is defined over the parameter set $w_k$ of the local model $\mathcal{M}_k$ and proxy model $\mathcal{Q}_k$ on client $k$. In each communication round, a total of $E = E_A + E_B$ local steps are conducted on client $k$ in two stages. The start of the local update at communication round $t$ is denoted as $tE + 0$, where the first local iteration of Stage I begins after the global semantic and decision anchors are received from the server. Stage I involves $E_A$ steps of local update, with parameter sequence denoted as $\{w_k^{tE+e}\}_{e=0}^{E_A}$, and $w_k^{tE+E_A}$ indicating the parameters at the end of this stage. Starting from $w_k^{tE+E_A}$, Stage II proceeds with $E_B$ local update steps, with parameters denoted by $\{w_k^{tE+e}\}_{e=E_A}^{E_A+E_B}$, where $w_k^{tE+E_A+E_B}$ represents the final parameters after the $t$-th local training round. After Stage II, the local representations and decision outputs are uploaded to the server for global aggregation and update, initiating the next communication round. The learning rates on client $k$ are unified into a scalar $\eta$, comprising $\eta_\psi$ and $\eta_\phi$.

**Assumption 1.** *Lipschitz Smoothness.The local loss function on any client $k$ is assumed to satisfy the 1-Lipschitz smoothness condition. Specifically, for any parameter vectors $w_k^{t_1}, w_k^{t_2}$, it holds that:*

$$\|\nabla \mathcal{L}_k^{t_1}(w_k^{t_1}; x, y) - \nabla \mathcal{L}_k^{t_2}(w_k^{t_2}; x, y)\| \le L_1 \|w_k^{t_1} - w_k^{t_2}\|, \forall t_1, t_2 > 0, (x, y) \in D_k \quad (22)$$

*Moreover, this can be further expressed as:*

$$\mathcal{L}_k^{t_1} - \mathcal{L}_k^{t_2} \le \langle \nabla \mathcal{L}_k^{t_2}, (w_k^{t_1} - w_k^{t_2}) \rangle + \frac{L_1}{2} \left\| w_k^{t_1} - w_k^{t_2} \right\|^2 \quad (23)$$

**Assumption 2.** *Unbiased Gradient and Bounded Variance.On client $k$, during the $t$-th local update, the batch gradient sampled at $w_k^t$ is denoted as $g_{w,k}^t = \nabla \mathcal{L}_k^t(w_k^t; B_k^t)$, and it is assumed to satisfy unbiasedness and bounded variance:*

*Unbiasedness:*

$$\mathbb{E}_{B_k^t \subseteq D_k}[g_{w,k}^t] = \nabla \mathcal{L}_k^t(w_k^t). \quad (24)$$

*Bounded variance:*

$$\mathbb{E}_{B_k^t \subseteq D_k} \left[ \left\| \nabla \mathcal{L}_k^t(w_k^t; B_k^t) - \nabla \mathcal{L}_k^t(w_k^t) \right\|^2 \right] \le \sigma^2 \quad (25)$$

**Assumption 3.** *Bounded Gradient in Global Alignment.Inspired by the theoretical bound on semantic alignment loss variation per round in FedProto, the first-stage local update iterations in FedKWAZ are defined, during which global semantic and decision alignment is performed on each client's local model. The resulting gradient shift from each single round is uniformly upper-bounded as:*

$$\|\nabla \mathcal{L}_{k,\text{Global}_{\text{align}}}^{tE+e}\|^2 \le \delta^2, \forall e \in \{0, 1, ..., E_A - 1\}, k \in \{0, 1, ..., N-1\}. \quad (26)$$

**Assumption 4.** *Bounded Gradient in KWAZ Alignment.Similarly, in the second stage of FedKWAZ (the KWAZ learning stage), the knowledge alignment operations (including SWAZ and DWAZ) performed between the local heterogeneous model and the proxy model on the client side are assumed to exhibit a unified bound on per-step gradient variation:*

$$\|\nabla \mathcal{L}_{k,\text{KWAZ}_{\text{align}}}^{tE+e}\|^2 \le \gamma^2, \forall e \in \{E_A, E_A + 1, ..., E_A + E_B - 1\}, k \in \{0, 1, ..., N-1\} \quad (27)$$

**Lemma 1.** *Stage-I Bias. Given Assumptions 1, 2 and 3, After the completion of the first stage, the following inequality is satisfied for any client:*

$$\mathbb{E}[\mathcal{L}_k(w_k^{tE+E_A})] \le \mathcal{L}_k(w_k^{tE+0}) - \left(\eta - \frac{L_1\eta^2}{2}\right) \sum_{e=0}^{E_A-1} \| \nabla \mathcal{L}_k(w_k^{tE+e}) \|_2^2 + \frac{L_1 E_A \eta^2 \sigma^2}{2} + \eta E_A \delta^2 \quad (28)$$

**Lemma 2.** *Stage-II Bias. Given Assumptions 1, 2 and 4, After the completion of the second stage, the following inequality is satisfied for any client:*

$$\mathbb{E}\left[\mathcal{L}_k(w_k^{tE+E_A+E_B})\right] \le \mathbb{E}\left[\mathcal{L}_k(w_k^{tE+E_A})\right] - \left(\eta - \frac{L_1\eta^2}{2}\right) \sum_{e=E_A}^{E_A+E_B-1} \|\nabla \mathcal{L}_k(w_k^{tE+e})\|_2^2$$
$$+ \frac{L_1 E_B \eta^2 \sigma^2}{2} + \eta E_B \gamma^2 \quad (29)$$

**Theorem 1.** *One Complete Round of FL. Given the above lemma, for any client, after the two-stage mutual learning has been completed, the following inequality holds:*

$$\mathbb{E}[\mathcal{L}_k(w_k^{tE+E_A+E_B})] \leq \mathcal{L}_k(w_k^{tE+0}) - \left(\eta - \frac{L_1\eta^2}{2}\right)\sum_{e=0}^{E_A+E_B-1} \| \nabla\mathcal{L}_k(w_k^{tE+e}) \|_2^2$$

$$+ \frac{L_1(E_A+E_B)\eta^2\sigma^2}{2} + \eta(E_A\delta^2 + E_B\gamma^2) \tag{30}$$

**Theorem 2.** *Non-convex Convergence Rate of FedKWAZ. Given Theorem 1, For any client and any constant $\varepsilon > 0$, the following inequality holds:*

$$\frac{1}{T}\sum_{t=0}^{T-1}\sum_{e=0}^{E_A+E_B-1} \|\nabla\mathcal{L}_k(w_k^{tE+e})\|_2^2$$

$$\leq \frac{\frac{1}{T}\sum_{t=0}^{T-1}\left[\mathcal{L}_k(w_k^{tE+0}) - \mathbb{E}[\mathcal{L}_k(w_k^{tE+E_A+E_B})]\right] + \frac{L_1(E_A+E_B)\eta^2\sigma^2}{2} + \eta(E_A\delta^2 + E_B\gamma^2)}{\eta - \frac{L_1\eta^2}{2}} < \epsilon$$

$$s.t. \quad 0 < \eta < \frac{2(\epsilon - (E_A\delta^2 + E_B\gamma^2))}{L_1(\epsilon + (E_A+E_B)\sigma^2)}. \tag{31}$$

## C.1  Proof of Lemma 1

*Proof.* In the $e$-th local update of the first stage ($e \in 0, 1, ..., E_A - 1$), any client $k$ is considered, and the result can be derived based on the Lipschitz smoothness in 1.

$$\mathbb{E}[\mathcal{L}_k(w_k^{tE+e+1})] \overset{(a)}{\leq} \mathbb{E}\left[\mathcal{L}_k(w_k^{tE+e}) + \langle\nabla\mathcal{L}_k(w_k^{tE+e}), w_k^{tE+e+1} - w_k^{tE+e}\rangle + \frac{L_1}{2}\|w_k^{tE+e+1} - w_k^{tE+e}\|_2^2\right]$$

$$\overset{(b)}{=} \mathcal{L}_k(w_k^{tE+e}) + \mathbb{E}\left[\langle\nabla\mathcal{L}_k(w_k^{tE+e}), -\eta g_k^{tE+e}\rangle + \frac{L_1}{2}\|\eta g_k^{tE+e}\|_2^2\right]$$

$$= \mathcal{L}_k(w_k^{tE+e}) - \eta\,\mathbb{E}\left[\langle\nabla\mathcal{L}_k(w_k^{tE+e}), g_k^{tE+e}\rangle\right] + \frac{L_1\eta^2}{2}\mathbb{E}\left[\|g_k^{tE+e}\|_2^2\right]$$

$$\overset{(c)}{=} \mathcal{L}_k(w_k^{tE+e}) - \eta\|\nabla\mathcal{L}_k(w_k^{tE+e})\|_2^2 + \frac{L_1\eta^2}{2}\mathbb{E}\left[\|g_k^{tE+e}\|_2^2\right]$$

$$\overset{(d)}{\leq} \mathcal{L}_k(w_k^{tE+e}) - \eta\|\nabla\mathcal{L}_k(w_k^{tE+e})\|_2^2 + \frac{L_1\eta^2}{2}\left(\|\nabla\mathcal{L}_k(w_k^{tE+e})\|_2^2 + \sigma^2\right)$$

$$= \mathcal{L}_k(w_k^{tE+e}) - \left(\eta - \frac{L_1\eta^2}{2}\right)\|\nabla\mathcal{L}_k(w_k^{tE+e})\|_2^2 + \frac{L_1\eta^2\sigma^2}{2}$$

Step (a): Based on Assumption 1, $\mathcal{L}_k$ is approximated near $w_k^{tE+e}$ using a first-order Taylor expansion.

Step (b): The update rule $w_k^{tE+e+1} = w_k^{tE+e} - \eta g_k^{tE+e}$ is applied, where $g_k^{tE+e}$ denotes the stochastic gradient over the current batch.

Step (c): Based on Assumption 2 (unbiased gradients).

Step (d): From Assumption 2 (bounded variance of the gradient), it holds that

$$\mathbb{E}[\| g_k^{tE+e} - \nabla\mathcal{L}_k(w_k^{tE+e}) \|_2^2] \leq \sigma^2 \tag{32}$$

From this, the following upper bound is derived:

$$\mathbb{E}[\| g_k^{tE+e} \|_2^2] \leq \| \nabla\mathcal{L}_k(w_k^{tE+e}) \|_2^2 + \sigma^2 \tag{33}$$

By summing the single-step inequalities from $e = 0$ to $e = E_A - 1$, the following result is obtained:

$$\mathbb{E}[\mathcal{L}_k(w_k^{tE+E_A})] \leq \mathcal{L}_k(w_k^{tE+0}) - \left(\eta - \frac{L_1\eta^2}{2}\right)\sum_{e=0}^{E_A-1} \| \nabla\mathcal{L}_k(w_k^{tE+e}) \|_2^2 + \frac{L_1E_A\eta^2\sigma^2}{2} \tag{34}$$

Furthermore, based on Assumption 3 (bounded global alignment gradient in the first stage), a per-round gradient variation upper bound $\delta^2$ is introduced, leading to an extra term $\eta E_A\delta^2$ in the total sum. Finally, Lemma 1 is fully expressed as Eq. 28. $\square$

## C.2 Proof of Lemma 2

*Proof.* Similarly, based on Assumptions 1, 2, and 4, after the second stage of FedKWAZ (KWAZ knowledge alignment stage), the expected local loss at any client $k$ is shown to satisfy Eq. 29. $\square$

## C.3 Proof of Theorem 1

*Proof.* By substituting Lemma 1 into the right-hand side of the inequality in Lemma 2, Eq. 30 can be obtained. $\square$

## C.4 Proof of Theorem 2

*Proof.* The left and right sides of Theorem 1 are swapped, and the gradient terms are rearranged:

$$\sum_{e=0}^{E_A+E_B-1} \parallel \nabla\mathcal{L}_k(w_k^{tE+e}) \parallel_2^2$$
$$\leq \frac{\mathcal{L}_k(w_k^{tE+0}) - \mathbb{E}[\mathcal{L}_k(w_k^{tE+E_A+E_B})] + \frac{L_1(E_A+E_B)\eta^2\sigma^2}{2} + \eta(E_A\delta^2 + E_B\gamma^2)}{\eta - \frac{L_1\eta^2}{2}}. \tag{35}$$

In the total $T$ rounds of federated communication training, the expectation of the above inequality is taken for $t$ from 0 to $T-1$ and summed, yielding:

$$\frac{1}{T}\sum_{i=0}^{T-1}\sum_{e=0}^{E_A+E_B-1} \parallel \nabla\mathcal{L}_k(w_k^{tE+e}) \parallel_2^2$$
$$\leq \frac{\frac{1}{T}\sum_{t=0}^{T-1}\left[\mathcal{L}_k(w_k^{tE+0}) - \mathbb{E}[\mathcal{L}_k(w_k^{tE+E_A+E_B})]\right] + \frac{L_1(E_A+E_B)\eta^2\sigma^2}{2} + \eta(E_A\delta^2 + E_B\gamma^2)}{\eta - \frac{L_1\eta^2}{2}}. \tag{36}$$

The difference between the loss at the initial time and the optimal loss is defined as:

$$\Delta = \mathcal{L}_0 - \mathcal{L}^* > 0 \tag{37}$$

The expected loss function over $T$ rounds is expressed as:

$$\frac{1}{T}\sum_{t=0}^{T-1}\left[\mathcal{L}_k(w_k^{t+0}) - E[\mathcal{L}_k(w_k^{tE+E_A+E_B})]\right] \leq \frac{\Delta}{T} \tag{38}$$

Hence, the original inequality can be further simplified as:

$$\frac{1}{T}\sum_{t=0}^{T-1}\sum_{e=0}^{E_A+E_B-1} \parallel \nabla\mathcal{L}_k(w_k^{tE+e}) \parallel_2^2 \leq \frac{\frac{\Delta}{T} + \frac{L_1(E_A+E_B)\eta^2\sigma^2}{2} + \eta(E_A\delta^2 + E_B\gamma^2)}{\eta - \frac{L_1\eta^2}{2}}. \tag{39}$$

Let the expected norm of the modulus in the above equation be expected to converge to a constant $\epsilon$:

$$\frac{\frac{\Delta}{T} + \frac{L_1(E_A+E_B)\eta^2\sigma^2}{2} + \eta(E_A\delta^2 + E_B\gamma^2)}{\eta - \frac{L_1\eta^2}{2}} < \epsilon \tag{40}$$

Since the number of training iterations $T > 0$, and $\Delta > 0$, the denominator must satisfy:

$$\epsilon(\eta - \frac{L_1\eta^2}{2}) - \frac{L_1(E_A+E_B)\eta^2\sigma^2}{2} - \eta(E_A\delta^2 + E_B\gamma^2) > 0 \tag{41}$$

Thus, the upper bound of $\eta$ is obtained as:

$$0 < \eta < \frac{2(\epsilon - (E_A\delta^2 + E_B\gamma^2))}{L_1(\epsilon + (E_A + E_B)\sigma^2)} \tag{42}$$

$\square$

Since all of the quantities $\epsilon$, $L_1$, $\sigma^2$, $\delta^2$, and $\gamma^2$ are positive finite constants, the constraint on the learning rate $\eta$ exists in a non-empty solution set. When the learning rate $\eta$ satisfies the above conditions, the expected norm of the local loss for any client can converge to the constant $\epsilon$. According to the right-hand side of the above equation, except for the term divided by $\frac{1}{T}$, the remaining terms are constants. Therefore, the non-convex convergence rate of FedKWAZ is achieved as: $O(1/T)$.

## D Additional Experimental Details

### D.1 Experimental Environment

All experiments are conducted on the PyTorch platform. The experiments are executed on four NVIDIA GeForce 4090 GPUs (24GB memory) across five supervised image classification datasets [2].

### D.2 Datasets

The sources of the datasets are detailed. The experiments are conducted based on five public multi-class datasets, covering natural and medical image recognition tasks, including:
Cifar10 (https://pytorch.org/vision/main/generated/torchvision.datasets.CIFAR10.html),
Cifar100 (https://pytorch.org/vision/stable/generated/torchvision.datasets.CIFAR100.html),
Flowers102 (https://pytorch.org/vision/stable/generated/torchvision.datasets.Flowers102.html),
Tiny-ImageNet (http://cs231n.stanford.edu/tiny-imagenet-200.zip),
and Skin-Lesions-14 (https://www.kaggle.com/datasets/ahmedxc4/skin-ds) are utilized.

### D.3 Hyperparameter Settings

In addition to the hyperparameter settings provided in the main text, the hyperparameter configurations for each baseline method are also followed according to their original publications. Specifically, LG-FedAvg is configured with no additional hyperparameters; for FedGen, the noise dimension is set to 32, the generator learning rate is set to 0.1, and the hidden dimension is aligned with the feature dimension $K$, with 100 training rounds on the server. The distillation hyperparameters in FML are set as $\alpha = 0.5$ and $\beta = 0.5$; in FedKD, the proxy model's learning rate is set to 0.01 to match the client; the temperature range for distillation is set to $T_{\text{start}} = 0.95$ and $T_{\text{end}} = 0.95$; for FedDistill, $\gamma = 1$ is applied. FedProto is configured with $\lambda = 0.1$; in FedGH, the server learning rate is set to 0.01 to match the client; the representation dimension of the proxy model in FedMRL is set to 256. For FedTGP, $\lambda$ is set to 0.1, the distillation margin threshold $\tau$ to 100, and the server training rounds to 100; in FedKTL, $K$ is set to $C$, $\mu = 50$, $\lambda = 1$, with server learning rate $\eta_s = 0.01$, batch size $B_s = 100$, and server training rounds $E_s = 100$. Except for FedGen and FedKTL, where server-side training is performed using the Adam [16] optimizer, SGD [46] is applied for client- and server-side training in all other methods. In FedKWAZ, the HAPM module is employed to automatically search for optimal mixing strength parameters ($\alpha$ or $\beta$) using KDP within the range $0.1, 0.5, 1.0$. The spatial granularity $G$ is used to divide each input image into $\sqrt{G} \times \sqrt{G}$ local patches. Specifically, for CIFAR10 and CIFAR100 (input size $3 \times 32 \times 32$), as well as Flowers102 and Tiny-ImageNet (input size $3 \times 64 \times 64$), $G \in 64, 16, 4$; for Skin-Lesions-14 (input size $3 \times 28 \times 28$), $G \in 49, 16, 4$. In FedKWAZ, the KWAZ update frequency (i.e., the interval $k$ for KDP-based HAPM parameter search) is fixed to once every 30 rounds to balance the ability to absorb prior knowledge and explore novel knowledge; the distillation temperature $\tau$ is set to 4. The training epoch $E_A$ and $E_B$ for the first and second stages are uniformly set to 1.

### D.4 Detailed Setting of Model Heterogeneity

The primary test configuration for feature extractor heterogeneity is defined as $\text{HtFE}_8$, consisting of eight representative network architectures: 4-layer CNN [21], GoogleNet [29], MobileNet_v2 [27], ResNet18, ResNet34, ResNet50, ResNet101, and ResNet152 [10]. Networks of ResNet18 and deeper are classified under the ResNet series and are used to construct multi-level heterogeneous configurations, including: $\text{HtFE}_1$ (containing only ResNet4, used to simulate a fully homogeneous scenario), $\text{HtFE}_2$ (composed of CNN and ResNet18, indicating slight heterogeneity), $\text{HtFE}_4$ (integrating CNN, GoogleNet, MobileNet_v2, and ResNet18 to simulate moderate heterogeneity), and

---
[2]Code is available at: https://github.com/ysml666/FedKWAZ

HtFE$_9$ (configured as extremely heterogeneous by combining ResNet4, ResNet6, ResNet8, ResNet10, ResNet18, ResNet34, ResNet50, ResNet101, and ResNet152).

Regarding classifier heterogeneity, the HtC$_4$ scenario is constructed to include four types of classifiers composed only of fully connected (FC) layers, to simulate structural heterogeneity in decision-making. The four architectures are defined as: (1) single-layer FC (100-d), (2) two-layer FC (512-d $\rightarrow$ 100-d) with a 512-d hidden layer, (3) two-layer FC (256-d $\rightarrow$ 100-d), and (4) two-layer FC (128-d $\rightarrow$ 100-d). In the HtFE$_8$-HtC$_4$ scenario, the eight types of feature extractors and four types of classifiers are cross-combined according to client indices, thereby forming a test environment with simultaneous heterogeneity in both feature extraction and decision modules.

In mutual learning schemes, the proxy model structure is defined as a simple 4-layer CNN to suit methods such as FedKD, FML, and FedMRL that periodically upload parameters of the proxy models to the server, aiming to reduce communication overhead. In contrast, FedKWAZ avoids proxy model transmission entirely and instead constructs global semantic prototypes and decision anchors by aggregating local class-wise features and logits, the proxy model is still uniformly set as a 4-layer CNN to ensure consistency and fairness in experimental comparisons, and to eliminate the influence of model complexity as a confounding variable.

## E  Additional Experimental Results

### E.1  Impact of Feature Dimensions

As shown in Table 5, most algorithms exhibit accuracy gains as the feature dimension increases from $K = 64$ to $K = 256$. At $K = 256$, FedKWAZ attains an accuracy of 50.95%, surpassing FedKTL by 5.16% (Figure 7), indicating its strong capability in leveraging high-dimensional representations for effective knowledge interaction. When the feature dimension is further increased to $K = 1024$, performance declines are observed in several methods, which can be attributed to elevated sparsity and reduced discriminative efficiency in overly expanded feature spaces. In contrast, FedKWAZ maintains competitive performance with 50.87% accuracy, supported by its KWAZ-guided knowledge coordination.

In addition, several federated learning baselines are outperformed by the Local method under certain configurations, as negative transfer under dual heterogeneity in data and model structures compromises collaborative training. FedKWAZ, by incorporating KWAZ-guided modeling of representation and decision discrepancies, effectively alleviates cross-model knowledge transfer bottlenecks and maintains consistent performance across diverse tasks and configurations.

Table 5: Impact of feature dimensions ($K$) on HtFE$_8$ model group performance on Cifar100.

|  | $K = 64$ | $K = 256$ | $K = 1024$ |
|---|---|---|---|
| Local | 39.24±0.14 | 40.92±0.11 | 40.25±0.18 |
| FedDistill | 38.93±0.20 | 44.10±0.15 | 42.56±0.23 |
| LG-FedAvg | 39.66±0.22 | 40.15±0.14 | 41.25±0.19 |
| FedGen | 38.75±0.15 | 40.23±0.19 | 40.43±0.11 |
| FedKD | 40.54±0.16 | 40.26±0.17 | 41.08±0.10 |
| FedProto | 31.84±0.10 | 35.63±0.12 | 34.14±0.14 |
| FML | 38.40±0.18 | 40.80±0.11 | 40.59±0.19 |
| FedGH | 38.19±0.18 | 40.01±0.15 | 38.48±0.17 |
| FedMRL | 39.06±0.14 | 41.90±0.16 | 42.95±0.12 |
| FedTGP | 47.05±0.17 | 47.87±0.27 | 47.43±0.24 |
| FedKTL | 45.98±0.15 | 45.79±0.14 | 46.76±0.17 |
| FedKWAZ | **50.04±0.13** | **50.95±0.16** | **50.87±0.12** |

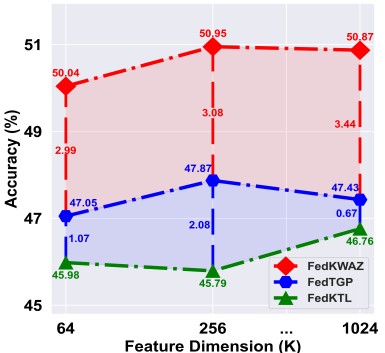

Figure 7:  Accuracy comparison among FedKWAZ, FedTGP, and FedKTL.

### E.2  Homogeneous Models Setting

To further investigate the role of data heterogeneity in isolation, all clients are configured with identical model architectures across three homogeneous settings: ResNet10, ResNet18, and ResNet34, in addition to the original HtFE$_1$ (ResNet4) baseline listed in Table 2. By removing model-level

heterogeneity, these experiments focus solely on the impact of non-IID data distributions on federated knowledge transfer. As shown in Table 6, FedKWAZ consistently outperforms all baseline methods under each homogeneous configuration. These results demonstrate that, even without structural differences across models, the proposed dual-stage mutual learning framework remains effective by explicitly aligning both representation semantics and decision behaviors across clients. This joint alignment enables FedKWAZ to bridge knowledge gaps induced purely by data distribution shifts, thereby ensuring stable performance improvements under non-IID conditions.

### E.3 Performance on Fashion-MNIST under Pathological and Dirichlet Data Settings

On the Fashion-MNIST [40] dataset, the $HtCNN_8$ model group is adopted to accommodate the grayscale single-channel input, with partition details listed in Table 8. A comprehensive evaluation of all methods is conducted under both Pathological and Practical non-IID settings. As shown in Table 7, FedKWAZ consistently yields the highest accuracy in both scenarios. Although FMNIST presents relatively modest classification complexity and most methods perform well, FedKWAZ maintains a consistent performance lead. In addition, Figure 8 presents the t-SNE [34] visualization of learned feature representations under the pathological setting for representative mutual learning baselines (FML, FedKD, FedMRL) and FedKWAZ. Compared to these baselines, the features produced by FedKWAZ exhibit stronger intra-class compactness and inter-class separability, reflecting enhanced semantic consistency under model heterogeneity and data distribution shift, and supporting improved cross-model knowledge transfer.

Table 6: Performance using homogeneous models on Cifar100 in the practical setting.

| Architectures | ResNet10 | ResNet18 | ResNet34 |
|---|---|---|---|
| Local | 45.38±0.16 | 42.57±0.12 | 41.62±0.14 |
| FedDistill | 44.78±0.18 | 44.12±0.21 | 43.62±0.23 |
| LG-FedAvg | 47.11±0.17 | 44.53±0.15 | 44.04±0.20 |
| FedGen | 47.02±0.22 | 44.53±0.13 | 44.27±0.11 |
| FedKD | 45.13±0.11 | 41.32±0.15 | 40.26±0.12 |
| FedProto | 40.67±0.25 | 40.23±0.18 | 38.02±0.15 |
| FML | 46.37±0.16 | 43.07±0.17 | 40.25±0.12 |
| FedGH | 45.30±0.12 | 43.29±0.14 | 41.84±0.10 |
| FedMRL | 47.36±0.09 | 45.67±0.10 | 45.40±0.12 |
| FedTGP | 47.05±0.38 | 45.79±0.35 | 47.43±0.33 |
| FedKTL | 51.20±0.15 | 50.10±0.14 | 48.17±0.12 |
| FedKWAZ | **52.35±0.11** | **52.47±0.14** | **51.75±0.12** |

Table 7: Performance on FMNIST using the $HtCNN_8$.

| Settings | Pathological Setting | Practical Setting |
|---|---|---|
| Local | 99.38±0.05 | 97.22±0.09 |
| FedDistill | 99.42±0.04 | 97.44±0.03 |
| LG-FedAvg | 99.37±0.05 | 97.22±0.04 |
| FedGen | 99.34±0.06 | 97.34±0.05 |
| FedKD | 99.40±0.07 | 97.36±0.03 |
| FedProto | 99.39±0.03 | 97.35±0.02 |
| FML | 99.41±0.05 | 97.34±0.04 |
| FedGH | 99.38±0.04 | 97.36±0.03 |
| FedMRL | 99.45±0.05 | 97.08±0.04 |
| FedTGP | 99.52±0.06 | 97.53±0.05 |
| FedKWAZ | **99.60±0.03** | **97.61±0.04** |

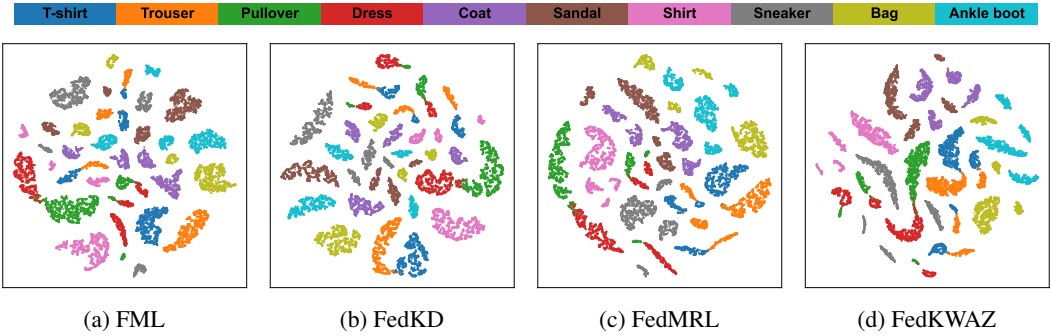

Figure 8: T-SNE visualization of features extracted by FML, FedKD, FedMRL and FedKWAZ on FMNIST under pathological partition.

### E.4 Performance under Low Client Participation Rates and High Client Drop Rates

On the Cifar10 dataset (Dirichlet distribution, 100 clients), client participation is systematically restricted to simulate federated scenarios with limited and unstable communication. Participation rates of 5% and 10% are adopted, corresponding to only 5 or 10 randomly selected clients contributing to each round, while aggregation uses updates only from the clients participating in that round. For FML, FedKD, and FedMRL, global proxy models are constructed by aggregating local proxy models

Table 8: The model architectures in the HtCNN$_8$ group. Convolutional layers are represented as "[$5 \times 5, 32$]" indicating a convolution with kernel size $5 \times 5$ and 32 output channels, and "$2 \times 2$ max pooling" refers to a max pooling layer with kernel size $2 \times 2$.

| | Sequentially Connected Feature Extractors | Classifiers |
|---|---|---|
| CNN1 | Conv ($5 \times 5$, 32), $2 \times 2$ max pool, 512-d fc | 10-d fc |
| CNN2 | Conv ($5 \times 5$, 32), $2 \times 2$ max pool, Conv ($5 \times 5$, 64), $2 \times 2$ max pool, 512-d fc | 10-d fc |
| CNN3 | Conv ($5 \times 5$, 32), $2 \times 2$ max pool, $2 \times$ 512-d fc | 10-d fc |
| CNN4 | Conv ($5 \times 5$, 32), $2 \times 2$ max pool, Conv ($5 \times 5$, 64), $2 \times 2$ max pool, $2 \times$ 512-d fc | 10-d fc |
| CNN5 | Conv ($5 \times 5$, 32), $2 \times 2$ max pool, 1024-d fc, 512-d fc | 10-d fc |
| CNN6 | Conv ($5 \times 5$, 32), $2 \times 2$ max pool, Conv ($5 \times 5$, 64), $2 \times 2$ max pool, 1024-d fc, 512-d fc | 10-d fc |
| CNN7 | Conv ($5 \times 5$, 32), $2 \times 2$ max pool, 1024-d fc $\times 2$, 512-d fc | 10-d fc |
| CNN8 | Conv ($5 \times 5$, 32), $2 \times 2$ max pool, Conv ($5 \times 5$, 64), $2 \times 2$ max pool, 1024-d fc, 512-d fc $\times 2$ | 10-d fc |

from all clients. In contrast, FedKWAZ performs global aggregation using class-wise prototypes and logits from all clients to form semantic representations and decision anchors. As reported in Table 9, overall performance degrades as participation rate decreases. Nonetheless, FedKWAZ consistently achieves the highest accuracy at both 5% and 10% settings, demonstrating strong resilience to client sparsity. Figure 9 further illustrates the prediction matrix at 10% participation, where FedKWAZ exhibits sharper diagonal confidence compared to other methods, indicating improved inter-class separation and stronger decision reliability under sparse client engagement.

Table 9: Performance of FML, FedKD, FedMRL and FedKWAZ under low client participation rates and high client drop rates.

| | Client Participation Rate | | Client Drop Rate | |
|---|---|---|---|---|
| | 5% | 10% | 90% | 95% |
| FML | 79.54±0.10 | 80.40±0.12 | 81.55±0.14 | 81.12±0.11 |
| FedKD | 79.41±0.11 | 79.67±0.10 | 80.66±0.16 | 80.48±0.13 |
| FedMRL | 80.92±0.12 | 81.08±0.15 | 81.65±0.13 | 81.39±0.10 |
| FedKWAZ | **82.85±0.09** | **83.71±0.11** | **84.37±0.12** | **83.93±0.13** |

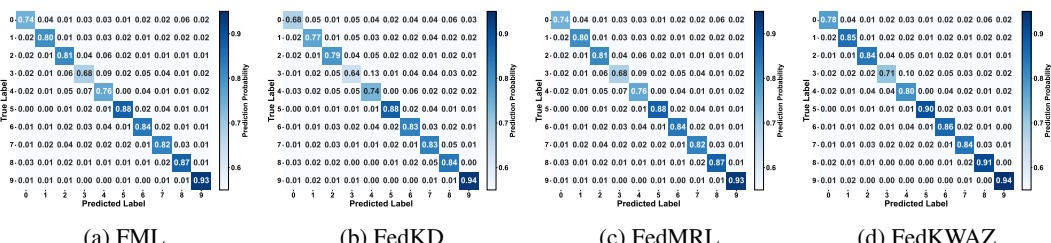

|          (a) FML          |          (b) FedKD          |          (c) FedMRL          |          (d) FedKWAZ          |

Figure 9: Confusion matrices of FML, FedKD, FedMRL, and FedKWAZ on Cifar10 under dirichlet partition with a client participation rate of 0.1, showing the prediction probability for each class.

To further evaluate communication disruptions, a Client Drop Rate experiment is conducted. In this setting, all 100 clients participate in local training, but only a portion are able to upload their knowledge due to simulated dropout. Dropout rates of 90% and 95% are used, meaning that only 10 or 5 clients successfully contribute to global aggregation. As shown in Table 9, these settings yield better accuracy than the 10% and 5% participation scenarios, suggesting that consistent local participation—even under partial upload failure—helps preserve training efficacy and partially compensates for reduced aggregation scale.

These results collectively confirm that FedKWAZ, empowered by its dual-stage knowledge transfer mechanism and lightweight semantic anchoring, maintains robust knowledge integration even under extreme communication sparsity and unstable client availability, effectively addressing practical challenges in real-world federated deployments.

## E.5   Impact of SWAZ and DWAZ

The individual contributions of SWAZ and DWAZ to the second-stage mutual learning in FedKWAZ are further investigated. Specifically, experiments are conducted by selectively enabling either SWAZ or DWAZ while disabling the other, and performance is assessed on the Skin-Lesions-14 and Flowers102 datasets under both Dirichlet and Pathological data partitions. As reported in Figure 10, DWAZ consistently achieves higher accuracy than SWAZ, indicating that decision-level alignment exerts a more immediate influence on cross-model knowledge transfer. This observation highlights the critical role of decision space consistency in federated mutual learning. Furthermore, the full FedKWAZ—combining both SWAZ and DWAZ—achieves the highest performance across all scenarios, confirming the complementary nature of semantic and decision-level discrepancy modeling. These findings underscore the necessity of jointly capturing both representational and predictive misalignments to optimize knowledge transfer between heterogeneous models.

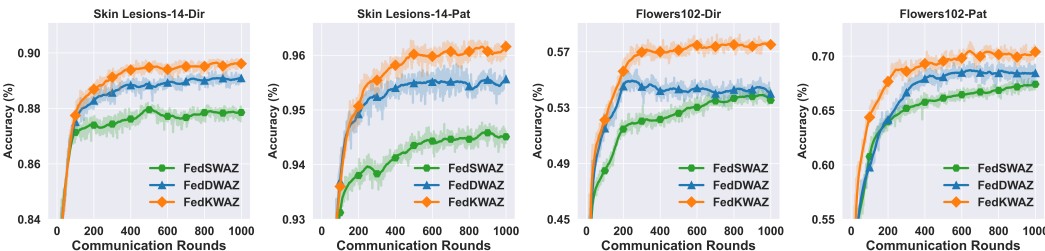

Figure 10: The contributions of SWAZ and DWAZ to performance.

## E.6   Impact of Feature Shift

To further assess the robustness of each method under feature shift scenarios, a heterogeneous data experiment is constructed in which each client is provided with a complete set of class labels, but the visual characteristics of the data—such as style, texture, and viewpoint—differ substantially across domains. Two widely adopted domain generalization benchmarks are selected: PACS [18], comprising 9,991 images from 7 categories across 4 domains (Art, Cartoon, Photo, Sketch), and OfficeHome [35], containing 15,588 images from 65 categories over 4 domains (Art, Clipart, Product, Real). For each dataset, samples are split into training and test sets with a 3:1 ratio, and each domain is treated as a distinct client, simulating federated feature heterogeneity.

In model allocation, the HtFE$_4$ architecture configuration is adopted, where heterogeneous backbones—including a 4-layer CNN, GoogleNet, MobileNet_v2, and ResNet18—are deployed across clients to emulate the hardware and model diversity typical in real-world federated environments. Figure 11 illustrates the domain-induced feature shift across different domains in PACS and OfficeHome for a representative category, highlighting the substantial variation in visual statistics. Table 10 summarizes the per-domain test accuracy at the final communication round and the overall average accuracy achieved at the optimal round for each method. Notably, the average accuracy is not obtained via direct averaging over individual client scores but is instead computed based on the total number of correct predictions across all test samples, yielding a more comprehensive and balanced measure of global performance.

As presented in Table 10 and Figure 12, FedKWAZ consistently demonstrates faster convergence, improved communication efficiency, and higher per-domain accuracies. In addition, it achieves the highest aggregated performance, significantly surpassing all baselines. These results suggest that FedKWAZ effectively addresses the representational and decision-level mismatches induced by feature shift through its dynamic KWAZ-based localization and targeted enhancement strategy, thereby enabling more reliable knowledge transfer across heterogeneous clients.

Table 10: Test accuracy (%) comparison of various federated learning methods on PACS and OfficeHome datasets. Each column represents the accuracy on one domain, while "Avg" indicates the mean accuracy across all domains.

| Methods | PACS | | | | | OfficeHome | | | | |
|---|---|---|---|---|---|---|---|---|---|---|
| | Art | Cartoon | Photo | Sketch | Avg | Art | Clipart | Product | Real | Avg |
| Local | 40.23±0.09 | 78.64±0.11 | 58.61±0.07 | 73.04±0.11 | 65.51±0.18 | 14.50±0.14 | 47.25±0.14 | 33.96±0.08 | 18.53±0.17 | 30.96±0.18 |
| FedDistill | 42.66±0.20 | 78.60±0.14 | 60.77±0.21 | 77.72±0.25 | 68.63±0.24 | 16.14±0.15 | 49.73±0.20 | 48.47±0.13 | 20.83±0.15 | 36.16±0.16 |
| LG-FedAvg | 40.04±0.14 | 76.62±0.23 | 60.77±0.23 | 74.47±0.19 | 65.79±0.15 | 14.00±0.19 | 46.43±0.13 | 37.84±0.10 | 19.08±0.12 | 31.57±0.08 |
| FedGen | 40.43±0.18 | 78.84±0.31 | 57.18±0.08 | 69.38±0.16 | 65.99±0.17 | 12.85±0.21 | 47.25±0.21 | 33.96±0.30 | 11.10±0.24 | 31.08±0.27 |
| FedKD | 42.40±0.08 | 78.50±0.21 | 62.44±0.23 | 76.16±0.15 | 67.03±0.16 | 16.31±0.18 | 45.60±0.08 | 36.76±0.16 | 18.62±0.17 | 30.18±0.15 |
| FedProto | 41.75±0.21 | 77.65±0.20 | 61.57±0.24 | 73.75±0.26 | 66.31±0.24 | 16.39±0.08 | 48.64±0.21 | 33.06±0.23 | 14.59±0.15 | 29.67±0.18 |
| FML | 42.21±0.18 | 78.67±0.08 | 59.57±0.10 | 71.92±0.22 | 65.11±0.15 | 15.17±0.13 | 43.95±0.26 | 33.34±0.19 | 17.42±0.23 | 30.83±0.14 |
| FedGH | 39.65±0.15 | 78.52±0.13 | 59.33±0.26 | 76.50±0.19 | 66.67±0.23 | 15.49±0.15 | 47.53±0.16 | 41.89±0.18 | 16.42±0.08 | 32.39±0.10 |
| FedMRL | 39.45±0.17 | 78.40±0.15 | 60.29±0.13 | 76.40±0.26 | 67.39±0.19 | 15.82±0.23 | 51.77±0.14 | 37.59±0.13 | 17.81±0.26 | 33.11±0.19 |
| FedTGP | 38.47±0.23 | 76.45±0.34 | 63.16±0.30 | 78.94±0.32 | 68.71±0.36 | 14.28±0.22 | 47.60±0.28 | 33.77±0.34 | 18.26±0.26 | 30.73±0.32 |
| FedKWAZ | 42.77±0.21 | 78.91±0.13 | 63.51±0.26 | 79.04±0.19 | 69.43±0.23 | 17.16±0.15 | 54.96±0.48 | 55.50±0.18 | 28.90±0.08 | 41.70±0.10 |

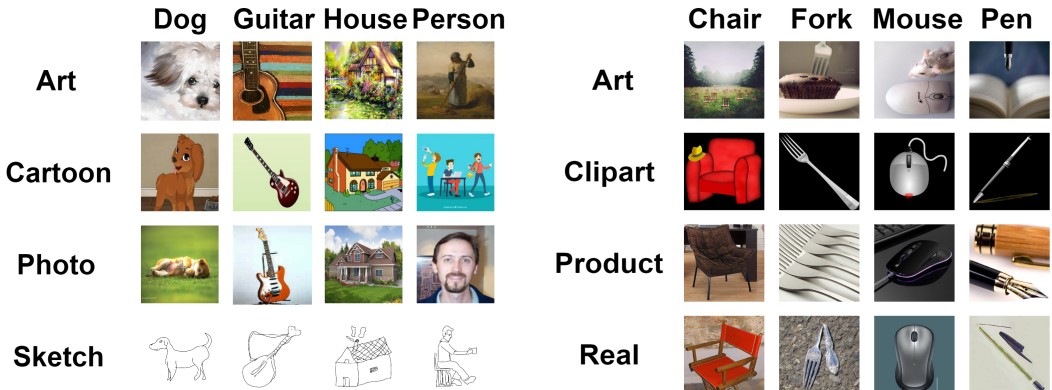

(a) Domain shift in PACS across four domains.  (b) Domain shift in OfficeHome across four domains.

Figure 11: Visualization of feature shift across different domains in PACS and OfficeHome datasets.

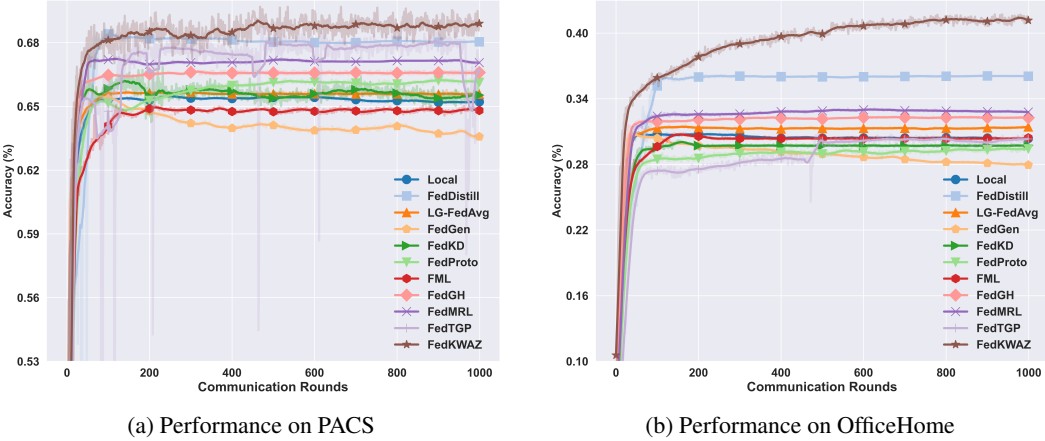

(a) Performance on PACS  (b) Performance on OfficeHome

Figure 12: Comparison of test accuracy across various methods on PACS and OfficeHome datasets.

### E.7 Effectiveness of HAPM vs. Hard Sample Selection

To further verify the effectiveness of the proposed Hierarchical Adaptive Patch Mixing (HAPM) module, we conducted additional comparative experiments to evaluate its advantage over naïve image distillation and local hard-sample selection strategies in facilitating cross-model knowledge transfer. The experiments were performed on two representative datasets, CIFAR-10 and Flowers102.

The compared methods include:

FedKD (Original): The private and proxy models align their features and logits using only original local images.

FedKD + Local Hard Sample Enhancement: The top 10%, 30%, and 50% local samples with the highest feature MSE and logits KL divergence were selected for enhanced distillation.

FedKWAZ (Full Framework): HAPM generates perturbed mixed samples, and KDP identifies semantic–decision weak zones (SWAZ and DWAZ) for fine-grained mutual learning.

Table 11: Performance comparison between FedKD, hard sample-enhanced FedKD (top 10%, 30%, 50% samples), and FedKWAZ on CIFAR-10 and Flowers102 datasets.

| Model | CIFAR-10 | Flowers102 |
|---|---|---|
| FedKD (Original) | 86.31 | 46.67 |
| FedKD + top 10% hard sample | 86.91 | 50.02 |
| FedKD + top 30% hard sample | 86.72 | 50.95 |
| FedKD + top 50% hard sample | 86.55 | 49.05 |
| FedKWAZ | **90.39** | **57.52** |

(1) As shown in Table 11., the performance gain from hard-sample enhancement exhibits dataset-dependent trends. On CIFAR-10, selecting the top 10% of hard samples yields the best result, while expanding the selection ratio slightly decreases performance. This indicates that the most challenging samples are highly informative, but excessive inclusion introduces redundancy and noise.

(2) On Flowers102, the best performance is achieved with the top 30% hard samples. Since this dataset has fewer training samples, an overly strict selection (e.g., 10%) limits generalization, whereas a moderate 30% ratio provides a better balance between informativeness and coverage.

(3) Overall, although selecting high-divergence samples can improve distillation, its effect is sensitive to the chosen ratio and lacks robustness. In contrast, FedKWAZ, equipped with HAPM for generating structurally diverse and perturbed samples and KDP for dynamically identifying discrepancy zones, consistently outperforms all hard-sample-based variants—achieving +4.08% gain on CIFAR-10 and +10.85% on Flowers102.

These results demonstrate that HAPM provides a more generalizable and stable mechanism for discrepancy exposure, eliminating the need for fixed selection thresholds and enabling models to better identify and assimilate knowledge weak-aware zones (KWAZ). Consequently, FedKWAZ supports more effective and fine-grained mutual distillation under heterogeneous federated learning settings.

### E.8 Correlation between HAPM Parameters and Client Properties

We investigate how the HAPM mixing parameters correlate with client properties. Eight clients (ID 0–7) are assigned private models in ascending capacity order: **4-layer CNN < MobileNet_V2 < GoogLeNet < ResNet18 < ResNet34 < ResNet50 < ResNet101 < ResNet152**. All clients use the same 4-layer CNN as the proxy model. On the **Flowers102** dataset (102 classes), we set the mixing strengths $\alpha, \beta \in \{0.1, 0.5, 1.0\}$, the spatial granularity $g \in \{4, 16, 64\}$, and the update frequency $k = 30$. Over 1000 rounds, 33 updates yield $8 \times 33 = 264$ HAPM configurations per client, each comprising $\{\text{SWAZ} : (\alpha^*, g^*)\}$ and $\{\text{DWAZ} : (\beta_1^*, g_1^*), (\beta_2^*, g_2^*)\}$.

Because KWAZ-guided mutual learning operates solely on local data, cross-client data heterogeneity has limited influence on **local parameter selection**; we therefore focus on **model architectural differences**. Based on the capacity gap between private and proxy models, clients are grouped into **Group A** (small gap, IDs 0–3; private models from 4-layer CNN to ResNet18) and **Group B** (large gap, IDs 4–7; private models from ResNet34 to ResNet152).

**Mixing strength ($\alpha, \beta$).** Smaller $\alpha$ or $\beta$ induces stronger information mixing, amplifying inter-model discrepancies. Empirically, $\alpha^*$, $\beta_1^*$, and $\beta_2^*$ most frequently take **0.1** (244, 233, and 220 times, respectively), indicating that KDP often prefers smaller mixing strengths to expose cross-model gaps. Smoother settings (0.5 or 1.0) occur **26 times in Group A** and **69 times in Group B**, showing that

when private and proxy models differ substantially, even smoother mixing can still form distinct discrepancy regions that benefit transfer and alignment.

**Spatial granularity** ($g$). A smaller $g$ produces fewer (larger) patches—coarser fusion—while a larger $g$ yields finer local mixing. The most frequent choice for $g^*$, $g_1^*$, and $g_2^*$ is **16** (238, 225, and 217 times, respectively), suggesting a medium granularity that balances discrepancy revelation and image recognizability. When $g = 4$, Group A appears **15 times** vs. Group B **38 times**; when $g = 64$, Group A appears **41 times** vs. Group B **18 times**. Thus, **larger structural gaps** favor **coarser mixing** (larger patch areas) to construct discrepancy regions, whereas **more similar models** benefit from **finer mixing** to explicitly stimulate subtle mismatches.

Overall, these results indicate that FedKWAZ exhibits **structure-aware behavior**: HAPM/KDP adaptively select mixing strengths and granularities according to the degree of architectural heterogeneity across clients.

### E.9 Variance and Stability Analysis of Dynamically Selected HAPM Parameters

To examine the dynamic behavior and stability of the HAPM parameters, we recorded the values of six key parameters ($\alpha^*, \beta_1^*, \beta_2^*, g^*, g_1^*, g_2^*$) selected by each client at every communication round and analyzed three representative rounds—the **30th**, **510th**, and **990th**—as summarized in Table 12.

Table 12: **Selected HAPM parameters across clients at rounds 30, 510, and 990.**

| Round | Client ID | $\alpha^*$ | $\beta_1^*$ | $\beta_2^*$ | $g^*$ | $g_1^*$ | $g_2^*$ |
|---|---|---|---|---|---|---|---|
| 30 | 0 | 0.1 | 0.1 | 0.1 | 16 | 16 | 16 |
| 30 | 1 | 0.1 | 0.1 | 0.1 | 16 | 64 | 16 |
| 30 | 2 | 0.1 | 0.1 | 0.1 | 16 | 16 | 16 |
| 30 | 3 | 0.1 | 0.5 | 0.1 | 16 | 4 | 16 |
| 30 | 4 | 0.1 | 0.1 | 0.1 | 16 | 16 | 16 |
| 30 | 5 | 0.1 | 0.1 | 0.5 | 16 | 16 | 64 |
| 30 | 6 | 0.1 | 1.0 | 0.1 | 16 | 16 | 16 |
| 30 | 7 | 0.5 | 0.1 | 0.1 | 4 | 16 | 16 |
| 510 | 0 | 0.1 | 0.1 | 0.1 | 16 | 16 | 64 |
| 510 | 1 | 0.1 | 0.1 | 0.1 | 16 | 16 | 16 |
| 510 | 2 | 0.1 | 0.1 | 0.1 | 16 | 64 | 16 |
| 510 | 3 | 0.1 | 0.1 | 0.5 | 16 | 16 | 16 |
| 510 | 4 | 0.1 | 0.1 | 0.1 | 16 | 16 | 16 |
| 510 | 5 | 0.1 | 0.1 | 0.5 | 64 | 16 | 16 |
| 510 | 6 | 1.0 | 0.1 | 0.1 | 16 | 4 | 16 |
| 510 | 7 | 0.1 | 0.5 | 0.1 | 16 | 16 | 4 |
| 990 | 0 | 0.1 | 0.1 | 0.1 | 16 | 16 | 16 |
| 990 | 1 | 0.1 | 0.1 | 0.1 | 16 | 16 | 16 |
| 990 | 2 | 0.1 | 0.1 | 0.1 | 16 | 16 | 16 |
| 990 | 3 | 0.1 | 1.0 | 0.1 | 4 | 16 | 16 |
| 990 | 4 | 0.1 | 0.1 | 0.1 | 16 | 16 | 16 |
| 990 | 5 | 0.1 | 0.1 | 0.1 | 16 | 16 | 4 |
| 990 | 6 | 0.5 | 0.1 | 0.1 | 16 | 4 | 16 |
| 990 | 7 | 0.1 | 0.1 | 0.5 | 16 | 16 | 16 |

**Inter-client variance.** In the **510th** round, the variance of $\alpha^*$ reached **0.089**, mainly because Client 6 (ResNet101, significantly more complex than the CNN proxy model) selected $\alpha^* = 1.0$. In such cases, clients with higher model capacities tend to prefer smoother mixing strategies to generate samples with stronger transferability. In contrast, in the **30th** round, the variance of $g_1^*$ reached **285.75**, primarily due to Client 1 (MobileNet_V2, a lightweight model) selecting $g_1^* = 64$, which is considerably higher than others. This observation suggests that when the private model is structurally close to the proxy model, the system adapts by employing finer-grained mixing strategies to better expose cross-model discrepancies.

**Intra-client variance.** Client 4 consistently selected identical values for all six parameters across rounds 30, 510, and 990, indicating strong internal stability. Similarly, most clients (Clients 0, 1, 2, and 5) maintained stable parameter selections across rounds, demonstrating that the KDP mechanism enforces a high degree of strategic consistency within each client.

Overall, FedKWAZ exhibits **low intra-client variance** (strong stability within clients) and **relatively high inter-client variance** (clear differentiation across clients). These findings confirm that FedKWAZ possesses a **structure-aware and adaptive KDP mechanism**, capable of dynamically adjusting mixing strategies according to model heterogeneity, thereby enhancing both robustness and transfer effectiveness in federated knowledge distillation.

# F  Visualizations of Data Distributions

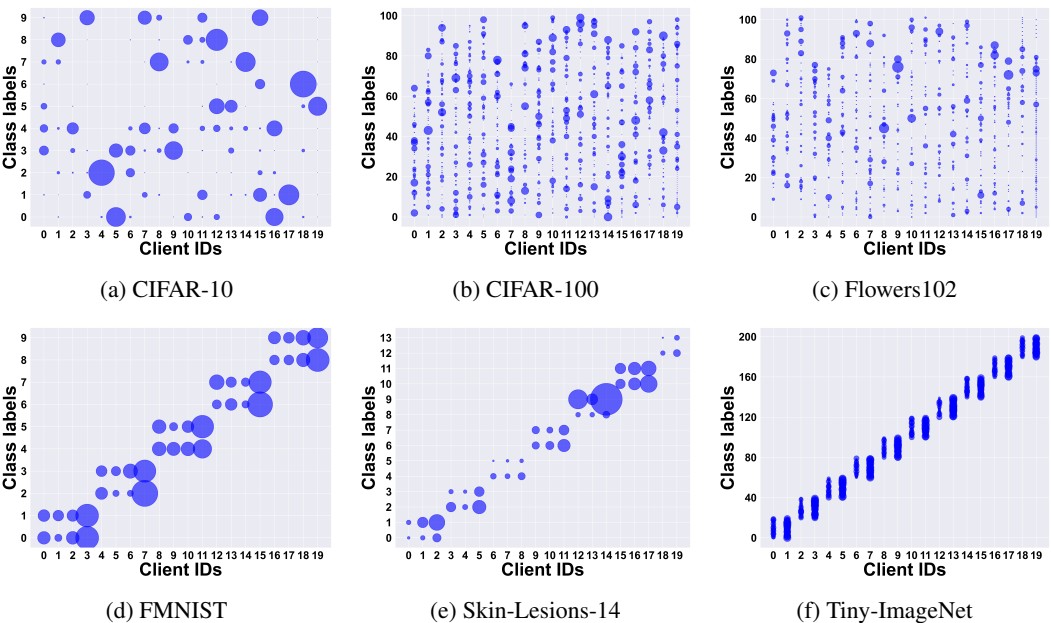

Figure 13: Top row (a–c) illustrates the **practical non-IID** scenario ($\beta = 0.1$) on CIFAR-10, CIFAR-100, and Flowers102. Bottom row (d–f) shows the **pathological non-IID** setting ($s = 2/2/20$) on FMNIST, Skin-Lesions-14, and Tiny-ImageNet. In each plot, each circle represents the class-wise data distribution of a client, and the size of the circle corresponds to the number of samples.

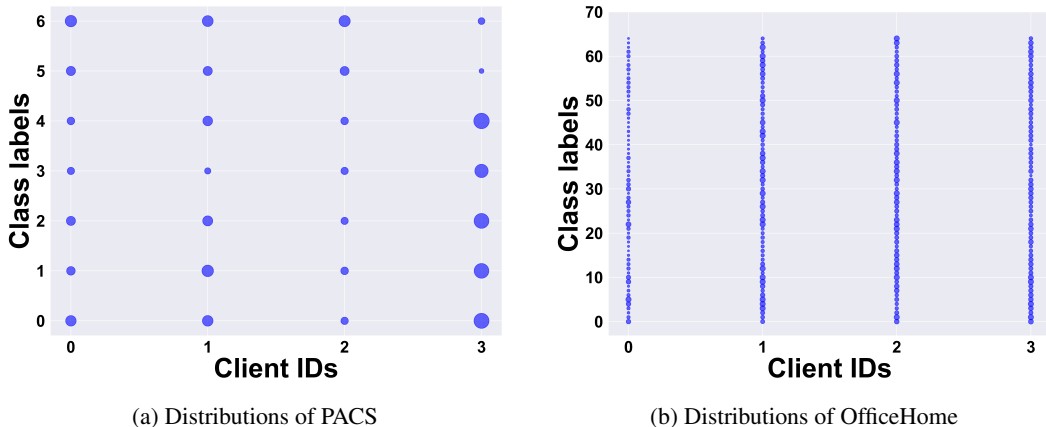

Figure 14: The data distributions of clients on PACS and OfficeHome, where each domain is assigned to an individual client. The size of each circle reflects the number of samples.

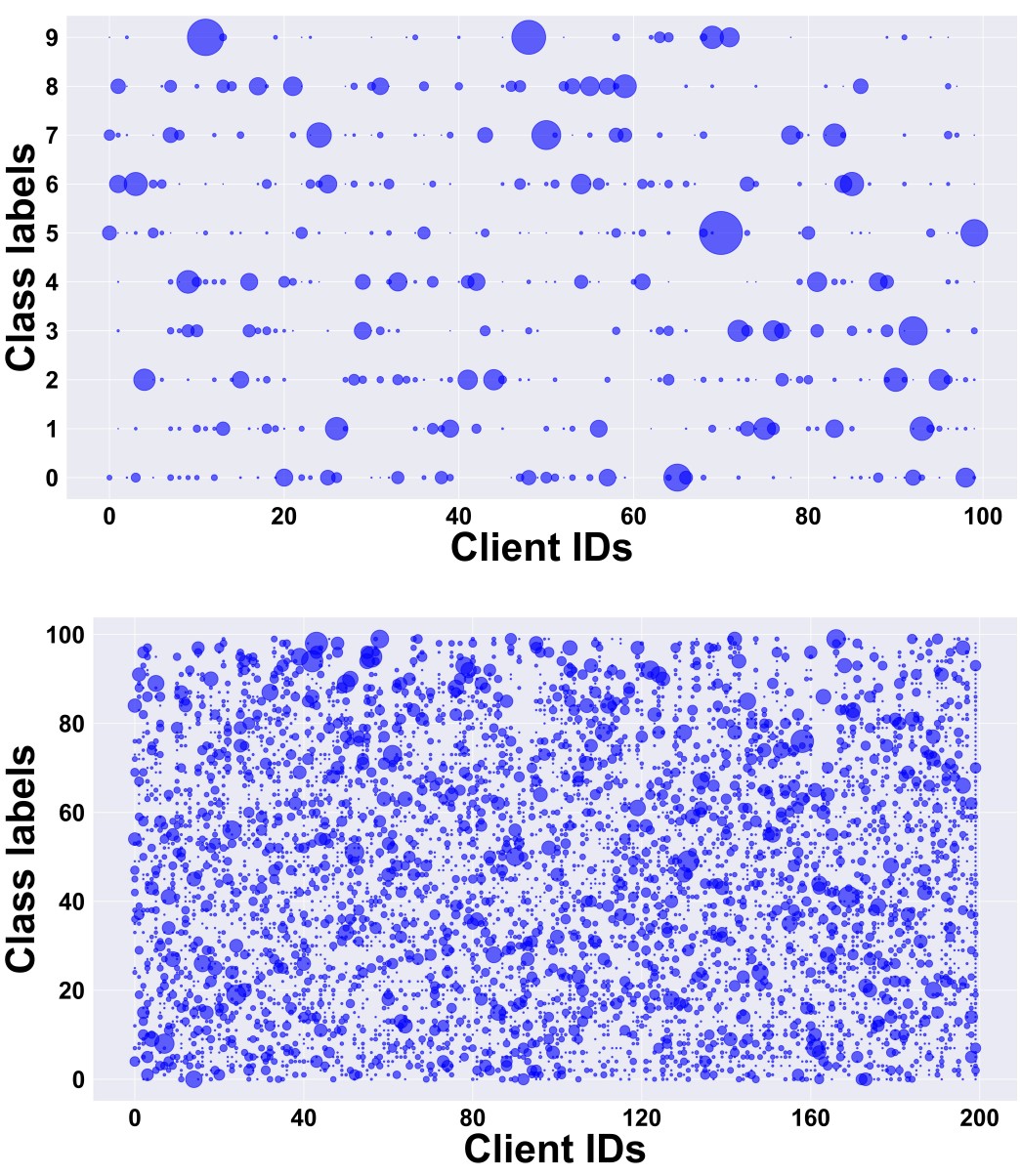

Figure 15: Data distributions of clients on Cifar10 and Cifar100 under practical settings ($\beta = 0.1$). The number of clients is 100 for Cifar10 and 200 for Cifar100, respectively. The size of each circle indicates the number of samples held by each client.

## G   Limitations

During the prediction distillation stage, a fixed temperature hyperparameter ($\tau = 4.0$) is employed. While the empirical results demonstrate consistent performance across heterogeneous model structures and data distributions, potential remains for optimizing distillation effectiveness by dynamically adjusting the temperature in response to model complexity or input data characteristics, which may further enhance system-level efficiency.

In the default experimental configuration, the feature dimension is uniformly set to $K = 512$. To assess the impact of dimensionality, additional experiments are conducted with $K = 64$, $256$, and $1024$ on the Cifar100 dataset. As datasets differ in their demands for feature expressiveness, determining feature dimensionality in accordance with data-specific properties warrants further investigation to improve representational capacity and adaptation across tasks.

## H   Broader Impacts

In the context of heterogeneous federated learning, a fine-grained mutual learning framework based on Knowledge Weak-Aware Zones (KWAZ) is proposed, yielding several broader impacts:

Enhanced adaptability of heterogeneous models. By explicitly modeling Semantic Weak-Aware Zones (SWAZ) and Decision Weak-Aware Zones (DWAZ), the proposed framework enables effective cross-architecture knowledge transfer. This design improves generalization under dual heterogeneity, supporting deployment in real-world scenarios with varying data distributions and model configurations.

Reduced communication overhead and improved privacy. By transmitting only class-level feature prototypes and prediction distributions, rather than full model parameters, the approach substantially reduces communication bandwidth consumption. Furthermore, as neither model structures nor weights are exposed, the system exhibits enhanced resistance to privacy leakage.

Improved system efficiency via faster convergence. The targeted refinement of KWAZ-guided distillation accelerates alignment in feature and decision spaces, enabling faster model convergence. This yields improved energy efficiency compared to conventional output-aligned distillation, contributing to the sustainability of large-scale distributed learning.

Broadened application potential. The architecture-agnostic design accommodates diverse client models and devices, promoting practical deployment in heterogeneous environments such as healthcare, smart manufacturing, and urban sensing systems. This broadens the real-world applicability of federated learning across domains.

