# OpenReview forum: "Transforming Gaps into Gains: Bridging Model and Data Heterogeneity in Federated Learning via Knowledge Weak-Aware Zones"
_NeurIPS.cc/2025/Conference — NeurIPS 2025 poster_

### Official Review · Reviewer_GNua · 2025-06-30

**Clarity:** 2
**Significance:** 3
**Originality:** 2
**Rating:** 4
**Confidence:** 3

**Summary:**

This paper introduces FedKWAZ, a novel mutual learning framework designed to overcome the shallow knowledge transfer common in heterogeneous federated learning (HtFL). The core contribution is the concept of "Knowledge Weak-Aware Zones" (KWAZ)—input regions where heterogeneous models disagree most, which are decomposed into Semantic (SWAZ) and Decision (DWAZ) zones to address discrepancies at both the representation and prediction levels. Using a Hierarchical Adaptive Patch Mixing (HAPM) module to create challenging samples and a Knowledge Discrepancy Perceptron (KDP) to dynamically identify these zones, FedKWAZ performs a targeted, KWAZ-guided local distillation. This is integrated into a two-stage process that starts with a communication-efficient global alignment using aggregated class-level anchors, and extensive experiments show that this approach significantly outperforms state-of-the-art HtFL methods.

**Questions:**

- Local Computational Cost: Could the authors provide a more detailed analysis of the local computational overhead introduced by the KDP search mechanism in Stage II? Specifically, is the search for the optimal pair exhaustive over the predefined sets A and G for every local training batch? A comparison of the wall-clock training time or FLOPS per local round against key baselines like FedTGP or FML would be very insightful to better understand the practical trade-offs between communication and computation.

- Can you show the actual value $\alpha$ that is selected per update round? I wonder if $\alpha$ is too high, the perturbed images do not represent class semantics but rather represent smoothed or generalized feature representation of the specific models.

- Regarding the distillation temperature (τ=4), this value appears relatively high compared to what is often used in standard knowledge distillation. Could the authors clarify the rationale behind this choice? It would be helpful to understand if this is a common practice for model-heterogeneous FL, or if there's a specific aspect of the FedKWAZ mechanism that benefits from a more pronounced softening of the probability distributions.

**Ethical Concerns:**

["NO or VERY MINOR ethics concerns only"]

**Final Justification:**

The authors addressed my concerns thoroughly with extensive empirical evidence.

**Limitations:**

Yes

**Quality:**

2

**Strengths And Weaknesses:**

### Strengths
- Focusing on discrepancy-focused alignment is technically solid and sound.

- The decomposition of KWAZ into semantic (SWAZ) and decision (DWAZ) components is logical, allowing for a fine-grained approach to correcting model mismatches at different levels of abstraction.

- The paper's experimental validation is thorough and rigorous. The evaluation spans multiple datasets (Cifar10/100, Flowers102, Skin-Lesions, Tiny-ImageNet) and also includes challenging domain-shift scenarios (PACS, OfficeHome). The methodical testing under various degrees of data (pathological, Dirichlet) and model (HtFE/HtC series) heterogeneity provides compelling evidence for the method's robustness and superiority.

- This paper provides a theoretical guarantee for the convergence of the proposed algorithm


### Weaknesses

- Clarity: The paper lacks comprehensive observations, motivations, and justification for the design choices of the proposed methods.
Specifically, 1) while the motivation to align the feature representations between the proxy model and the private model, unlike previous algorithms that focus on aligning only the logits, is clear, it lacks observations or comprehensive analysis of why the logit alignment in mutual learning is not enough for model heterogeneous FL. The lack of such analysis causes a lack of motivation for using HAPM, which uses a form of mixed, perturbed synthetic images. 2) Authors should discuss the design choice of using 2-stage local training, not using all the proposed regularization losses jointly. 3) The paper lacks the justification or motivation for making smoothed, perturbed synthetic images, not just selecting local hard images that incur a large information mismatch between the proxy model and the private model. The paper also lacks an extensive analysis of how HAPM resolves the information misalignment. Since there would be a trade-off because this hard regularization can prevent effective local optimization of the private model, which has a more complex structure, aligning feature representations between the proxy model and the private model does not always improve the performance. Therefore more detailed and comprehensive discussion of the proposed method will improve the quality of the paper.


- Ablation study of more local epochs and lower participation rates would validate the robustness of the proposed methods to the data and model heterogeneity in FL.

- Ablation study of using mutual learning losses without HAPM (i.e., using naive local images or selecting hard examples) will validate the effectiveness of using HAPM.

If the authors address by concerns, particularly regarding the clarity, I would be inclined to increase my rating.

---

> ### Author Rebuttal · Authors · 2025-07-30
>
> **W1**: We sincerely thank the reviewers for the suggestions on writing clarity.
>
> **On the limitation of “logits-only” alignment in existing methods**: the proxy and private models often differ in both their feature extractors and classifiers. Such heterogeneity leads not only to decision divergence at the output (logits) level, but also to semantic mismatch in the hidden feature space. As shown in Fig. 1, even with the same input, their embeddings differ significantly due to varied capacity in feature modeling. When passed through their own classifiers, this difference further **amplifies into prediction gaps**. Thus, methods that align only logits (e.g., FML) are difficult to establish a stable **feature-decision mapping** relationship due to the **accumulation of biases during the semantic representation stage**, thus limiting the effectiveness and consistency of knowledge transfer between models.
>
> In our results, FedKD—which aligns both features and logits—usually outperforms FML on complex datasets, showing the importance of feature alignment in addressing model-structure gaps.
>
> However, these methods still align features or logits directly on raw images, without identifying the key areas that drive cross-model gaps—resulting in shallow alignment. FedKWAZ tackles this by modeling the **“semantic–decision dual shift.”** It uses HAPM to generate local, perturbed samples with diverse semantics and KDP to locate KWAZ. Dual-path distillation is then applied on SWAZ and DWAZ to enable adaptive and targeted alignment, effectively going beyond prior FL distillation schemes.
>
> **Design Choice of the Two-Stage Local Training Scheme**: The decision to employ a two-stage local training strategy stems from the need to **balance global consistency and local discrepancy alignment effectively**.
>
>     Stage I-Global Anchoring: During this stage, the proxy and private models align each other's global class prototypes and logits after completing one round of training. This coarse-grained alignment helps establish a shared semantic space and basic decision boundaries, ensuring a stable foundation for subsequent alignment.
>
>     Stage II-KWAZ-Guided Distillation: Building on the preliminary alignment from Stage I, the second stage shifts its focus to learning the discrepancies within KWAZ. These regions highlight the most significant areas where the models differ. The goal in this stage is to address fine-grained differences in both semantic and decision-level spaces that were not fully resolved in the first stage.
>
> This design is motivated by the observation that combining all regularization losses simultaneously would **force the models to balance both global and local knowledge alignment at once**, risking model confusion and suboptimal learning. The two-stage approach ensures that each phase targets alignment at the appropriate granularity, optimizing the training process.
>
> **Regarding the use of HAPM-generated synthetic samples instead of directly selecting local hard samples**: Our adoption of the HAPM-based synthesis strategy is motivated by two main considerations.
>
> First, relying solely on local hard samples—i.e., those with large prediction gaps—cannot fully capture the diverse knowledge gap between private and proxy models. These samples are often few and **biased by the local data, making it hard to reveal broad differences in representations**. In contrast, HAPM introduces block-wise spatial mixing and local perturbation to produce diverse samples with mixed semantics. This leads to more informative contrastive signals and better exposure of representation or decision gaps between models.
>
> Second, the locally perturbed samples from HAPM bring mild **adversarial effects**, which help improve model robustness and generalization, while reducing overfitting risks from a few extreme samples.
>
> To address the reviewer’s concern about lacking analysis of how HAPM mitigates information mismatch, we clarify: the key role of HAPM is to break the agreement between proxy and private models on original samples by injecting localized noise and semantic mixing. This magnifies their representation or prediction differences, which are then modeled through contrastive loss. As a result, model alignment becomes more targeted and effective, systematically reducing information inconsistency caused by misaligned knowledge. Put differently, **HAPM is not only a generator, but also a divergence amplifier that enhances KDP’s ability to detect and correct gaps**.
>
> We also acknowledge the reviewer’s concern that overly strong alignment may harm local optima. To mitigate this, our design does not apply KWAZ-based distillation at every round. Instead, KDP is triggered periodically to dynamically locate areas of divergence, striking **a balance between consolidating existing knowledge and correcting emerging discrepancies**. In addition, **the mixing strength and patch granularity in HAPM are carefully controlled** to preserve semantic consistency and ensure the generated samples remain task-relevant. We also reorganized the introduction and related work to better present comprehensive observations, motivations, and justification for our design choices, as suggested by Reviewer c2QV.
>
> **W2**: We appreciate the reviewer’s suggestion regarding the choice of local training epochs and client participation rates. These factors indeed intensify bias and instability in federated learning, and we find the feedback highly valuable.
>
> Table 2: Test accuracy (%) on Cifar100 in the practical setting with large E.
>
> | Method    | E=5 | E=10 | E=20
> |----------|:--------:|:----------:|:----------:
> | FML   |  39.33   |   38.25    |   35.87
> | FedKD   |  41.28   |   40.25    |   38.75
> | FedMRL   |  42.41   |   42.97    |   43.65
> | FedTGP   |  46.82   |   46.36    |   47.03
> | FedKTL   |  46.87   |   46.62    |   45.98
> | FedKWAZ  | **51.76** | **51.97** |   **52.13**
>
> As shown in **Table 1**, we extended the number of local training epochs to 5, 10, and 20. Benefiting from the local **perturbation signals provided by HAPM and the mutual enhancement enabled by KWAZ**, FedKWAZ consistently improves as the number of local steps increases. Under low participation scenarios, FedKWAZ also maintains strong performance.
>
> Table 2: Test accuracy (%) on Tiny-ImageNet with 1000 clients under low client participation rates (ρ = 0.05).
>
> | Method    | $\rho=0.05$ |
> |----------|:----------:|
> | FedMRL   |   12.87    |
> | FedTGP   |   9.86    |
> | FedKTL   |   10.73    |
> | FedKWAZ  | **14.92** |
>
>  Furthermore, as shown in **Table 2**, we scaled the evaluation to 1000 clients on Tiny-ImageNet with only 5% participation, a setting characterized by extreme data sparsity. FedKWAZ achieved 14.92% accuracy under this condition, significantly outperforming FedKTL (10.73%) and FedMRL (13.42%). These results collectively demonstrate the robustness of FedKWAZ in handling both data and model heterogeneity. We will incorporate the full results and discussion into the revised manuscript.
>
> **W3**: We thank the reviewer for highlighting the need to validate the effect of HAPM. FedKD, which aligns features and logits using only original images, serves as a baseline. It underperforms FedKWAZ on multiple datasets, **showing that lacking a discrepancy-aware design limits its distillation power**.
>
> We further compared with a "local hard sample" strategy: selecting the top 10%, 30%, and 50% samples by logits gap. This method shows clear limits, as it uses few samples, is sensitive to bias, and tends to focus on narrow semantic areas. **It fails to reflect full model differences and may lead to overfitting**.
>
> By contrast, HAPM creates diverse samples via spatial perturbation and layered mixing. KDP then finds regions of high model disagreement. This setup better exposes semantic–decision shifts and allows for more adaptive and fine-grained distillation.
>
> **Q1**: We thank the reviewer for asking about the cost of HAPM and KDP. These steps are not done every round, but once per fixed interval (30 rounds by default). Each client runs forward propagation over a few mixing settings to find the one that maximizes model divergence. No backward propagation or weight updates are used, so the cost stays low.
>
> On CIFAR-100, KWAZ search adds only $1.58 \times 10^{3}$ GFlops per round on average, which is **8.93%** of total cost per round. Although FedKWAZ uses more GFlops per round ($1.77 \times 10^{4}$) than FML ($1.01 \times 10^{4}$) and FedTGP ($1.19 \times 10^{4}$), it trains **faster and reduces total compute and communication**. It also boosts accuracy by 12.22% over FML and 4.54% over FedTGP. We believe this trade-off is both fair and worthwhile.
>
> **Q2**: We thank the reviewer for pointing out the setting of the mixing factor $\alpha$. To balance semantic fidelity and perturbation strength, we restrict $\alpha$ to ${0.1, 0.5, 1.0}$. Lower values (e.g., 0.1, 0.5) yield stronger contrast in mixed areas, which helps expose local model differences. A larger $\alpha=1.0$ gives smoother blends that improve generalization.
>
> We observe that on complex datasets like CIFAR-100 and Tiny-ImageNet, KDP tends to pick smaller $\alpha$, showing that it prefers preserving semantic structure. Also, since HAPM mixes locally in image space, **even a large $\alpha$ only affects part of the input, not the global view**. This ensures a good trade-off between diversity and semantic retention.
>
> **Q3**: We also thank the reviewer for highlighting the choice of distillation temperature $\tau=4$. In heterogeneous FL, logits across models often vary sharply. In FedKWAZ, we target KWAZ areas with strong model disagreement, where predictions are more spread. If $\tau$ is too low, alignment may bias toward the dominant class and ignore minor classes. **A higher $\tau=4$ smooths the output and better balances all class signals, supporting fine-grained transfer**. This setting has shown consistent benefits in our experiments.

---

> > ### Comment · Reviewer_GNua · 2025-08-02
> > **Additional Concerns**
> >
> > I appreciate the authors' efforts in presenting a detailed rebuttal, but I have additional questions.
> >
> > 1. Assumptions 3, 4 assume that local gradients for each stage are bounded, which are quite strong and unrealistic assumptions. Upon this assumption, the rigorous convergence analysis of the proposed method in the presence of data heterogeneity is not possible since gradients of all clients will be bound to the identical value without any divergence term. To address data heterogeneity in convergence analysis, previous FL works usually assume that the difference between the norm of gradient for global loss and the average of norm for gradients of local clients is bounded (please refer to A2 in SCAFFOLD [1]), and do not assume the norm of gradient of each local loss is bounded.
> >
> > [1] S. P. Karimireddy, et al.: SCAFFOLD: Stochastic Controlled Averaging for Federated Learning, in ICML, 2020.
> >
> > 2. Can you provide empirical evidence (even simplified) to support the claims for the response W3?
> >
> > 3. What is the typical distribution of the selected values for $\alpha$, $\beta$, and $g$ across the experiments? Furthermore, is there an observable correlation between specific client properties (e.g., the degree of data heterogeneity, model architecture) and the parameters selected? For instance, do clients with more significant data skew or simpler models tend to prefer higher mixing strengths or a specific level of patch granularity?
> >
> > 4.  **Variance Analysis:** How significant is the variance of these dynamically chosen parameters? Specifically, could you elaborate on the degree of:
> >     *   **Inter-client variance:** How much do the selected values differ between clients within the same communication round?
> >     *   **Intra-client variance:** How stable are the selected values for a single client across different communication rounds?
> >
> > I would be very grateful if you could provide answers to the questions above. Understanding these aspects would provide deeper insights into the adaptability and stability of the proposed framework.

---

> > > ### Author Response · Authors · 2025-08-07
> > > **Response to concern 1 (1/2)**
> > >
> > > We sincerely thank you for your valuable suggestions. Your comments have provided important directions for improvement and helped us further enhance the quality of our work.
> > >
> > > In response to your concerns regarding Assumptions 3 and 4, we conducted a more rigorous re-analysis of the gradient convergence in the two-stage mutual learning process. Following the assumption framework of FedProto, we retain the original Assumption 1 (Lipschitz smoothness) and Assumption 2 (unbiased gradient with bounded variance). To meet the needs of the new analysis, we introduce the following assumptions:
> > >
> > > Assumption 3 (Bounded gradient expectation): The expected gradient norm of the private model ($\psi$) and the proxy model ($\phi$) on client k is bounded: $
> > > \mathbb{E}[\\|g\_{\psi,k}\^t\\|\_2] \leq G\_1, \quad \mathbb{E}[\\|g\_{\phi,k}\^t\\|\_2] \leq G\_2,
> > > $
> > >
> > > Assumption 4 (Lipschitz continuity of the embedding function): Each local embedding function is L2-Lipschitz continuous: $
> > > \\|\mathcal{F}(x;\omega\_k\^{t\_1}) - \mathcal{F}(x;\omega\_k\^{t\_2})\\|\_2 \leq L\_2 \\|\omega\_k\^{t\_1} - \omega\_k\^{t\_2}\\|
> > > $
> > >
> > > Assumption 5 (Lipschitz continuity of the decision function): Each local decision function is L3-Lipschitz continuous: $\|\mathcal{H}(\mathcal{F}(x;\omega_k^{t_1});\omega_k^{t_1})-\mathcal{H}(\mathcal{F}(x;\omega_k^{t_2});\omega_k^{t_2})\|_2\leq L_3\|\omega_k^{t_1}-\omega_k^{t_2}\|$
> > >
> > > Assumption 6 (Lipschitz continuity of the cross-entropy loss): The cross-entropy loss is L4-Lipschitz continuous with respect to the predicted logits: $
> > > |\ell\_{\mathrm{CE}}(\hat{y}\^{t\_1}, q) - \ell\_{\mathrm{CE}}(\hat{y}\^{t\_2}, q)| \leq L\_4 \\|\hat{y}\^{t\_1} - \hat{y}\^{t\_2}\\|\_2
> > > $
> > >
> > > Based on these assumptions, we analyze the gradient change of the private model learning from the proxy model direction in two consecutive rounds. Considering the symmetry of the mutual learning process, we combine the gradient changes of the private model learning from the proxy and the proxy model learning from the private, and take the expectation to obtain the complete gradient for one communication round. The detailed derivation is as follows:
> > >
> > > **Stage I: Global prototype alignment**:
> > > $
> > > \Delta\mathcal{L}\_{\mathrm{MSE},e}\^{\mathcal{M}\to\mathcal{Q}} = \\|\bar{P}\_{c}\^{\mathcal{Q},t} - z\_{x}\^{\mathcal{M}\_{k},tE+e+1}\\|\_2 - \\|\bar{P}\_{c}\^{\mathcal{Q},t} - z\_{x}\^{\mathcal{M}\_{k},tE+e}\\|\_2
> > > \overset{(a)}{\leq} \\|z\_{x}\^{\mathcal{M}\_{k},tE+e+1} - z\_{x}\^{\mathcal{M}\_{k},tE+e}\\|\_2
> > > \overset{(b)}{\leq} L\_2 \\|\psi\_{k}\^{tE+e+1} - \psi\_{k}\^{tE+e}\\|\_2
> > > = L\_2 \eta \\|g\_{\psi,k}\^{tE+e}\\|\_2
> > > \overset{(c)}{\leq} L\_2 \eta G\_1
> > > $
> > >
> > > where (a) follows $ \\|a - b\\|\_2 - \\|a - c\\|\_2 \leq \\|b - c\\|\_2 $, (b) follows from Assumption 4, (c) follows from Assumption 3.
> > >
> > > **Global logits alignment**:
> > >
> > > $
> > > \Delta \mathcal{L}\_{\text{CE},e}\^{\mathcal{M} \to \mathcal{Q}} = \ell\_{\text{CE}}(\hat{y}\_x\^{\mathcal{M}\_k, tE+e+1}, \sigma(\bar{L}\_c\^{\mathcal{Q}, t})) - \ell\_{\text{CE}}(\hat{y}\_x\^{\mathcal{M}\_k, tE+e}, \sigma(\bar{L}\_c\^{\mathcal{Q}, t}))
> > > \overset{(a)}{\leq} L\_4 \\| \hat{y}\_x\^{\mathcal{M}\_k, tE+e+1} - \hat{y}\_x\^{\mathcal{M}\_k, tE+e} \\|\_2
> > > $
> > > $
> > > {\leq} L\_4 \\| \mathcal{H}(\mathcal{F}(x; \psi\_k\^{tE+e+1}); \psi\_k\^{tE+e+1}) - \mathcal{H}(\mathcal{F}(x; \psi\_k\^{tE+e}); \psi\_k\^{tE+e}) \\|\_2
> > > \overset{(b)}{\leq} L\_4 L\_3 \\| \psi\_k\^{tE+e+1} - \psi\_k\^{tE+e} \\|\_2
> > > = L\_4 L\_3 \eta \\| g\_{\psi,k}\^{tE+e} \\|\_2
> > > \overset{(c)}{\leq} L\_3 L\_4 \eta G\_1,
> > > $
> > >
> > > where (a), (b), and (c) follow from Assumptions 6, 5, and 3, respectively.

---

> > > ### Author Response · Authors · 2025-08-07
> > > **Response to concern 2**
> > >
> > > We sincerely thank the reviewer for their attention to our response in W3 and for pointing out the need to provide more empirical evidence supporting the effectiveness of HAPM. This valuable feedback motivated us to design and conduct a set of additional studies to verify, from an empirical perspective, the superiority of HAPM over **naive image distillation and local hard-sample strategies in facilitating knowledge transfer**.
> > >
> > > To this end, we conducted the following comparative experiments on two representative datasets, CIFAR-10 and Flowers102:
> > >
> > >     FedKD (Original): The private and proxy models align their features and logits using only original local images.
> > >
> > >     FedKD + Local Hard Sample Enhancement: The top 10%, 30%, and 50% local samples with the highest feature MSE and logits KL divergence were selected for enhanced distillation.
> > >
> > >     FedKWAZ (Full Framework): HAPM generates perturbed mixed samples, and KDP identifies the semantic–decision weak zones for fine-grained mutual learning.
> > >
> > > Table 1: Performance comparison between FedKD, hard sample-enhanced FedKD (top 10%, 30%, 50% samples), and FedKWAZ on CIFAR-10 and Flowers102 datasets.
> > >
> > > | Model    | Cifar10 | Flowers102 |
> > > |----------|:--------:|:----------:|
> > > | FedKD (Original)   |  86.31   |   46.67    |
> > > | FedKD + top 10% hard sample  |  86.91   |   50.02    |
> > > | FedKD + top 30% hard sample   |  86.72   |   50.95    |
> > > | FedKD + top 50% hard sample   |  86.55   |   49.05    |
> > > | FedKWAZ  | **90.39** | **57.52** |
> > >
> > >
> > > (1) As shown in **Table 1**, the performance gain from hard sample enhancement exhibits distinct trends across datasets. On CIFAR-10, selecting the top 10% of hard samples yields the highest gain. As the selection ratio increases, the improvement slightly decreases. This suggests that, due to the abundance of CIFAR-10 data, the most extreme hard samples are highly informative, while enlarging the selection range may introduce noise and redundancy. Thus, "focusing on a very small number of the hardest examples" proves most effective.
> > >
> > > (2) On Flowers102, the best performance is observed when using the top 30% hard samples. We speculate this is because Flowers102 has significantly fewer training samples than CIFAR-10. An overly strict selection (e.g., 10%) leads to too few examples for generalization, while a 30% ratio provides a better trade-off between informativeness and coverage without overfitting or bias.
> > >
> > > (3) The overall trend from both datasets leads to a consistent conclusion: although selecting high-divergence samples for distillation offers some performance gain, **it is sensitive to the chosen ratio and has limited robustness**. In contrast, FedKWAZ, with HAPM generating structurally diverse and perturbed images, and KDP dynamically locating discrepancy zones, consistently outperforms all hard-sample-based approaches—achieving +4.08% gain on CIFAR-10 and +10.85% on Flowers102.
> > >
> > > This comparison highlights that HAPM provides a more **generalizable and stable mechanism for discrepancy exposure, avoiding the reliance on fixed thresholds and enabling the models to better recognize and absorb KWAZ**—thus supporting more effective and fine-grained distillation. We greatly appreciate the reviewer’s attention and insightful guidance on this important point.

---

> ### Author Response · Authors · 2025-08-07
> **Response to concern 1 (2/2)**
>
> **Stage II: Local prototype alignment**:
>
> $\Delta\_\text{{MSE}}\^{\mathcal{M} \to \mathcal{Q}} = \\| z\_x\^{\mathcal{M}\_k, tE+e+1} - z\_x\^{\mathcal{Q}\_k, tE+e+1} \\|\_2\^2 - \\| z\_x\^{\mathcal{M}\_k, tE+e} - z\_x\^{\mathcal{Q}\_k, tE+e} \\|\_2\^2 = \\| (z\_x\^{\mathcal{M}\_k, tE+e+1} - z\_x\^{\mathcal{M}\_k, tE+e}) - (z\_x\^{\mathcal{Q}\_k, tE+e+1} - z\_x\^{\mathcal{Q}\_k, tE+e}) \\|\_2\^2$
>
> $+ 2 \langle (z\_x\^{\mathcal{M}\_k, tE+e+1} - z\_x\^{\mathcal{M}\_k, tE+e}) - (z\_x\^{\mathcal{Q}\_k, tE+e+1} - z\_x\^{\mathcal{Q}\_k, tE+e}), z\_x\^{\mathcal{M}\_k, tE+e} - z\_x\^{\mathcal{Q}\_k, tE+e} \rangle \overset{(a)}{\leq} \\| (z\_x\^{\mathcal{M}\_k, tE+e+1} - z\_x\^{\mathcal{M}\_k, tE+e}) - (z\_x\^{\mathcal{Q}\_k, tE+e+1} - z\_x\^{\mathcal{Q}\_k, tE+e}) \\|\_2\^2 $
>
> $\overset{(b)}{\leq} [L\_2 \eta (\\| g\_{\psi,k}\^{tE+e} \\|\_2 + \\| g\_{\phi,k}\^{tE+e} \\|\_2)]\^2 \overset{(c)}{\leq} L\_2\^2 \eta\^2 (\\| g\_{\psi,k}\^{tE+e} \\|\_2 + \\| g\_{\phi,k}\^{tE+e} \\|\_2)\^2 \overset{(d)}{\leq} L\_2\^2 \eta\^2 (G\_1 + G\_2)\^2$
>
> where (a) follows from expanding the squared difference, with a focus on the dominant quadratic term between feature updates, (b) follows from Assumption 4, (c) follows squaring the bound and maintaining the linear term, (d) follows from Assumption 3.
>
> **Local logits alignment**:
>
> $\Delta \mathcal{L}\_{\text{KL},e}\^{\mathcal{M} \to \mathcal{Q}} = \text{KL}\left( \sigma\left(\hat{y}\_x\^{\mathcal{M}\_k,tE+e+1} / \tau \right) \middle\| \middle\| \sigma\left(\hat{y}\_x\^{\mathcal{Q}\_k,tE+e+1} / \tau \right) \right) - \text{KL}\left( \sigma\left(\hat{y}\_x\^{\mathcal{M}\_k,tE+e} / \tau \right) \middle\| \middle\| \sigma\left(\hat{y}\_x\^{\mathcal{Q}\_k,tE+e} / \tau \right) \right) $
>
> $\overset{(a)}{\leq} \left\langle \nabla\_{\hat{y}} \text{KL}, \Delta \hat{y}\^{\mathcal{M}\_k} - \Delta \hat{y}\^{\mathcal{Q}\_k} \right\rangle \overset{(b)}{\leq} L\_3 \cdot \\|\Delta \hat{y}\^{\mathcal{M}\_k} - \Delta \hat{y}\^{\mathcal{Q}\_k} \\|\_2 \overset{(c)}{\leq} L\_3 \cdot (L\_3 \eta (G\_1 + G\_2)) = L\_3\^2 \eta (G\_1 + G\_2)$
>
> where (a) follows from the first-order Taylor expansion of the KL divergence, (b) follows from Assumption 5, (c) follows from Assumption 3 and 5.
>
> **Through the above analysis, we obtain the gradient change of the private model learning from the proxy model in a single round of mutual learning. The gradient of the proxy model learning from the private model can be derived symmetrically**.
> By combining the gradient contributions of two consecutive rounds (i.e., round e and e+1) and taking the expectation, the gradient bound for one complete communication round can be obtained. We recognize that the current theoretical framework still requires further refinement, and we will continue working to improve it.

---

> ### Author Response · Authors · 2025-08-07
> **Response to concern 3**
>
> We thank the reviewer for suggesting the exploration of potential correlations between the mixing parameters in FedKWAZ and client properties. We assigned the models in the $\text{HtFE}_8$ pool as private models for 8 clients (ID 0–7) according to their capacity in **ascending order** (4-layer CNN < MobileNet_V2 < GoogLeNet < ResNet18 < ResNet34 < ResNet50 < ResNet101 < ResNet152). All clients used the same 4-layer CNN as the proxy model. On the Flowers102 dataset with 102 complex semantic classes, we set the mixing strength parameter $\alpha$ ∈ {0.1, 0.5, 1.0}, and the spatial granularity parameter $g$ ∈ {4, 16, 64}, and used the default update frequency (k = 30). Over 1000 rounds of training, a total of 33 updates were performed, **generating 8×33 = 264 groups of HAPM parameter configurations, where each group includes {SWAZ: ($\alpha^\star$, $g^\star$)} and {DWAZ: ($\beta_1^\star$, $g_1^\star$), ($\beta_2^\star$, $g_2^\star$)}**. Since the KWAZ mutual learning process is entirely based on the client's local data, data heterogeneity **(mainly reflected in distribution differences across clients)** has little impact on local parameter selection. Therefore, the focus turns to the influence of model architectural differences. **Based on the capacity difference between private and proxy models**, the 8 clients were divided into two groups: **Group A (small model gap, IDs 0–3, private models from 4-layer CNN to ResNet18) and Group B (large model gap, IDs 4–7, private models from ResNet34 to ResNet152)**, in order to analyze the impact of architectural heterogeneity on parameter selection.
>
> First, we examine the mixing strength parameters ($\alpha$ and $\beta$). Theoretically, smaller values of $\alpha$ or $\beta$ lead to stronger information mixing, which helps amplify knowledge differences between models. Statistics show that $\alpha^\star$, $\beta_1^\star$, and $\beta_2^\star$ were most frequently chosen as 0.1 **(244, 233, and 220 times respectively)**, indicating that KDP often selects smaller $\alpha$ or $\beta$ to reveal cross-model knowledge gaps. To further verify the influence of model structural difference on parameter selection, we found that the smoother **values (0.5 or 1.0) appeared 26 times in Group A and 69 times in Group B, showing that when private and proxy models are significantly different, even smoother mixing can generate distinct knowledge difference regions in the image, thus facilitating effective knowledge transfer and alignment**.
>
> Next, we analyze the selection of spatial granularity parameter $g$. In HAPM, a smaller $g$ means the image is divided into fewer patch regions, resulting in larger fusion areas (coarser granularity), while a larger $g$ corresponds to finer local mixing. The statistics show that $g^\star$, $g_1^\star$, and $g_2^\star$ most frequently took the value 16 **(238, 225, and 217 times respectively)**, indicating that medium granularity provides a better balance between revealing knowledge differences and preserving image recognizability. Further statistics show that when $g$=4, Group A appeared 15 times and Group B 38 times; when $g$=64, Group A appeared 41 times and Group B 18 times. This result reveals that when the private and proxy models have large structural differences, using coarser mixing (larger patch areas) can effectively construct knowledge difference regions, whereas when the models are structurally similar, finer mixing is needed to explicitly stimulate inter-model knowledge discrepancies. **This trend further confirms that FedKWAZ possesses structural difference awareness, enabling it to dynamically adapt mixing strategies according to the degree of heterogeneity**. We again thank the reviewer for raising this point, which provided an important opportunity for us to deepen our analysis of FedKWAZ’s strategy selection dynamics in heterogeneous settings.

---

> ### Author Response · Authors · 2025-08-07
> **Response to concern 4**
>
> We sincerely thank the reviewer for raising the important question regarding the dynamic behavior and variance analysis of HAPM parameters.
>
> Specifically, we recorded the values of key parameters ($\alpha^\star$, $\beta_1^\star$, $\beta_2^\star$, $g^\star$, $g_1^\star$, $g_2^\star$) selected by each client at every communication round, and focused on **three representative rounds (the 30th, 510th, and 990th)**, as shown in Table 1.
>
> Table 1: Selected HAPM parameters across clients at rounds 30, 510, and 990.
>
> | Round | Client ID | $\alpha^\star$ | $\beta_1^\star$ | $\beta_2^\star$ | $g^\star$ | $g_1^\star$ | $g_2^\star$ |
> |-------|-----------|----------------|------------------|------------------|-----------|-------------|-------------|
> | 30    | 0         | 0.1            | 0.1              | 0.1              | 16        | 16          | 16          |
> | 30    | 1         | 0.1            | 0.1              | 0.1              | 16        | 64          | 16          |
> | 30    | 2         | 0.1            | 0.1              | 0.1              | 16        | 16          | 16          |
> | 30    | 3         | 0.1            | 0.5              | 0.1              | 16        | 4           | 16          |
> | 30    | 4         | 0.1            | 0.1              | 0.1              | 16        | 16          | 16          |
> | 30    | 5         | 0.1            | 0.1              | 0.5              | 16        | 16          | 64          |
> | 30    | 6         | 0.1            | 1.0              | 0.1              | 16        | 16          | 16          |
> | 30    | 7         | 0.5            | 0.1              | 0.1              | 4         | 16          | 16          |
> | 510   | 0         | 0.1            | 0.1              | 0.1              | 16        | 16          | 64          |
> | 510   | 1         | 0.1            | 0.1              | 0.1              | 16        | 16          | 16          |
> | 510   | 2         | 0.1            | 0.1              | 0.1              | 16        | 64          | 16          |
> | 510   | 3         | 0.1            | 0.1              | 0.5              | 16        | 16          | 16          |
> | 510   | 4         | 0.1            | 0.1              | 0.1              | 16        | 16          | 16          |
> | 510   | 5         | 0.1            | 0.1              | 0.5              | 64        | 16          | 16          |
> | 510   | 6         | 1.0            | 0.1              | 0.1              | 16        | 4           | 16          |
> | 510   | 7         | 0.1            | 0.5              | 0.1              | 16        | 16          | 4           |
> | 990   | 0         | 0.1            | 0.1              | 0.1              | 16        | 16          | 16          |
> | 990   | 1         | 0.1            | 0.1              | 0.1              | 16        | 16          | 16          |
> | 990   | 2         | 0.1            | 0.1              | 0.1              | 16        | 16          | 16          |
> | 990   | 3         | 0.1            | 1.0              | 0.1              | 4         | 16          | 16          |
> | 990   | 4         | 0.1            | 0.1              | 0.1              | 16        | 16          | 16          |
> | 990   | 5         | 0.1            | 0.1              | 0.1              | 16        | 16          | 4           |
> | 990   | 6         | 0.5            | 0.1              | 0.1              | 16        | 4           | 16          |
> | 990   | 7         | 0.1            | 0.1              | 0.5              | 16        | 16          | 16          |
>
> **Inter-client variance**:
>
> In the 510th round, the variance of $\alpha^\star$ reached 0.089, due to Client 6 (ResNet101, significantly more complex than the CNN proxy model) selecting $\alpha^\star$ = 1.0. In such specific cases, clients with more complex models may opt for smoother mixing strategies to generate samples with better transferability.
>
> In the 30th round, the variance of $g_1^\star$ reached 285.75, mainly because Client 1 (MobileNet_V2, a lightweight model) selected $g_1^\star$ = 64, which is significantly higher than others. This suggests that when the private model is close in structure to the proxy model, the system may adapt finer-grained mixing strategies to better expose knowledge differences.
>
> **Intra-client variance**:
>
> Client 4 selected exactly the same values for all six parameters across rounds 30, 510, and 990, showing strong consistency.
>
> Most clients (Clients 0, 1, 2, and 5) also maintained stable parameter selections across rounds, indicating that the KDP ensures a high degree of strategic consistency within each client.
>
> Overall, FedKWAZ exhibits low intra-client variance (great stability within clients) and relatively high inter-client variance (differentiation across clients), which further validates the effectiveness and robustness of the proposed KDP mechanism in structure-aware and adaptive mixing strategy design. We once again thank the reviewer for raising this critical point.

---

> > ### Comment · Reviewer_GNua · 2025-08-09
> >
> > Thank you for your thorough and thoughtful response. Your detailed clarifications have addressed most of my concerns. In particular, the ablation study with hard negative samples and the analysis of parameter selection distributions were very helpful for deepening my understanding of the method, and I hope these can be incorporated into the final manuscript.
> >
> > Based on the addressed concerns, I will raise my rating to 4: Borderline Accept, to reflect my view that the paper meets the acceptance threshold for NeurIPS.
> >
> > Once again, thank you for the kind and detailed reply.
> >
> > Sincerely,

---

> > > ### Author Response · Authors · 2025-08-09
> > >
> > > We sincerely thank you for your positive feedback, valuable suggestions, and for raising the rating. We commit to incorporating the discussed analyses and improvements into the final manuscript. We truly appreciate your thoughtful review and kind support!

---

### Official Review · Reviewer_c2QV · 2025-07-03

**Clarity:** 2
**Significance:** 3
**Originality:** 3
**Rating:** 4
**Confidence:** 3

**Summary:**

This paper proposes FedKWAZ, a new framework for tackling heterogeneous federated learning (HtFL) problems where both data distributions and model architectures vary across clients. The key idea is to explicitly identify and exploit Knowledge Weak-Aware Zones (KWAZ). The authors introduce HAPM (Hierarchical Adaptive Patch Mixing) to generate diverse local perturbations and KDP (Knowledge Discrepancy Perceptron) to quantify feature and decision discrepancies, decomposing them into SWAZ (semantic) and DWAZ (decision) zones. The approach further combines a two-stage learning procedure with global prototype and decision anchor aggregation. Extensive experiments across multiple datasets, non-IID partitions, and heterogeneous model settings show significant improvements over prior SOTA methods.

**Questions:**

In the appendix, the authors analyze the convergence rate of the proposed algorithm as $O(1/T)$. However, at line 517, it appears that the authors directly assume the gradient norm is bounded by a constant $\epsilon$. This assumption is rather strong and not commonly justified in distributed training literature. Generally, existing works derive an inequality showing the gradient norm converges to zero as $T$ increases, from which the convergence rate is established. For example, the convergence rate of FedAvg is usually shown to be $O(1/\sqrt{T})$ [1].

[1] Tackling the Objective Inconsistency Problem in Heterogeneous Federated Optimization. NIPS 2020.

**Ethical Concerns:**

["NO or VERY MINOR ethics concerns only"]

**Final Justification:**

The authors addressed my question and concern. I'll keep my positive rating.

**Limitations:**

yes

**Quality:**

3

**Strengths And Weaknesses:**

##### Strengths

1. The paper directly targets the dual challenge of data and architectural heterogeneity in federated learning, which is highly relevant for practical deployments.
2. By maintaining private and proxy models on each client and using class-level prototypes and decision anchors for global alignment, the method avoids structural constraints of FedAvg and reduces communication.
3. Experiments cover diverse datasets (including medical images), multiple non-IID splits, and different architectural combinations, with ablations that isolate the roles of local mutual learning, KWAZ directionality, and SWAZ/DWAZ component, demonstrating the superior performance of the proposed method.

##### Weaknesses

1. Related work discussion is somewhat unbalanced. While the paper organizes related work into *Resource-, Modular-, and Autonomy-heterogeneous methods*, the discussion of Resource and Modular heterogeneity is relatively shallow compared to the extensive treatment of Autonomy-heterogeneous and mutual learning methods. It would strengthen the positioning to analyze existing works from the perspectives of *semantic representation alignment* and *decision distribution alignment*, which directly relate to the core contributions.
2. Core motivation could be clearer. The main novelty — using HAPM to find maximal representation/decision discrepancies (KWAZ), and then enforcing targeted bidirectional distillation — is sound, but the writing does not highlight this flow sufficiently. As a result, the paper sometimes feels like it lacks a clear driving question, especially in the related work section, which can confuse the reader about how exactly this method addresses gaps in prior approaches.

---

> ### Author Rebuttal · Authors · 2025-07-30
>
> **W1**: We thank the reviewer for suggesting the perspective of analyzing existing works through the lens of semantic representation alignment and decision distribution alignment. In the original submission, we categorized related works into three types: resource heterogeneity, modular heterogeneity, and autonomous heterogeneity, aiming to systematically summarize heterogeneity issues in federated learning and corresponding solutions. However, as the reviewer rightly pointed out, our analysis of the first two categories was relatively brief. We fully agree with this assessment and have accordingly enhanced the revised version by systematically **strengthening the technical analysis of resource- and modular-heterogeneous methods from both the semantic and decision alignment dimensions**.
>
> **W2**: We also thank the reviewer for highlighting the need to present the core motivation and conceptual pipeline more clearly. In the revised version, we have further strengthened the explanation of our methodological innovations to help readers better understand the driving problem addressed in this work and our proposed solution. Specifically, we explicitly formulate the core challenge as **"semantic–decision dual drift,"** which characterizes the systematic mismatch between heterogeneous models in both representation and decision space. This challenge is now clearly stated in the introduction as the central motivation of the paper.
> To address it, we clarify our complete strategy, which involves using HAPM + KDP to identify KWAZ regions and performing targeted bidirectional distillation over these critical areas. With these revisions, the research motivation and conceptual design are now more clearly presented, highlighting both the novel contribution of our method to deep knowledge alignment and its targeted response to limitations of prior work.
>
> Based on the reviewer's insightful suggestions and our analyses in W1 and W2, we have rewritten the second and third paragraphs of the Introduction. Additionally, we have revised the discussions on resource heterogeneity and architectural heterogeneity in the Related Work section, and restructured the concluding paragraph of that section accordingly.
>
> **Introduction**:
>
> (Paragraph 2): To tackle model and data heterogeneity in HtFL, federated mutual learning has been proposed, where clients train performance-driven private models and resource-friendly proxy models in parallel, facilitating knowledge transfer via bidirectional distillation. However, existing methods primarily focus on **direct alignment in the feature and decision space across heterogeneous models, overlooking deeper mismatches in their learned representations and decision behaviors**. As illustrated in Figure 1 (left), for the same input, the private and proxy models on a given client often produce significantly different feature distributions and classification outputs—a phenomenon termed **semantic–decision dual drift**. This dual drift implies that naive output matching yields only coarse and superficial knowledge transfer, leaving critical representation- and decision-level discrepancies unresolved. Without pinpointing and addressing these misaligned “weak zones,” knowledge transfer remains inefficient and ultimately constrains the client’s performance.
>
> (Paragraph 3): Motivated by the necessity to precisely address this semantic–decision dual drift，this paper proposes Knowledge Weak-Aware Zones (KWAZ), characterizing zones where heterogeneous models exhibit the strongest cognitive discrepancies. KWAZ is further decomposed into Semantic Weak-Aware Zones (SWAZ) and Decision Weak-Aware Zones (DWAZ), focusing respectively on representation and decision discrepancies that indicate key zones requiring targeted knowledge enhancement. Building on this insight, the FedKWAZ framework is designed to mine and align complementary knowledge within KWAZ through a two-stage mutual learning scheme integrating local and global collaboration.
>
> **Related Work**:
>
> **Resource-Heterogeneous FL Methods**. A number of approaches handle device heterogeneity by allowing clients to train smaller or pruned versions of a global model that fit their local resource constraints. For example, FedRolex, FjORD,FLASH,FedResCuE and HeteroFL enable each client to work with a sub-model or width-reduced network tailored to its hardware capabilities. These methods **focus on resource adaptation** — e.g., dynamically extracting or training sub-networks from a larger global model — so that both high-end and low-end devices can participate in federated training. However, their emphasis is on resource allocation **rather than on maintaining cross-client knowledge consistency**. As a result, resource-heterogeneous solutions do not explicitly align the semantic representations or decision distributions learned by different client models. The absence of such alignment means that knowledge gaps can persist between clients’ sub-models, **potentially limiting the overall knowledge transfer efficiency across the federation**.
>
> **Modular-Heterogeneous FL Methods**. Another line of work addresses model heterogeneity by splitting models into shared global modules and private local modules. FedRep and FedPer share and aggregates a global feature extractor across clients (aiming for aligned semantic representations), whereas LG-FedAvg and FedGH share a global classifier across clients (aiming for aligned decision output distributions). By partitioning the model, these approaches attempt to transfer knowledge either at the representation level (feature extractor) or at the decision level (classifier) among clients. However, since the **“global” module in such frameworks is obtained by aggregating components trained on each client’s non-iid local data**, forcing all clients to adopt the aggregated module can undermine personalization and lead to suboptimal local performance. In practice, these methods face **a trade-off between achieving global alignment and preserving local adaptation**. Merely sharing or exchanging modules may prove inadequate in addressing fine-grained, data-dependent misalignments in the learned representations or decision boundaries of client models, potentially leaving certain semantic or decision-level discrepancies unresolved.
>
> (The revised final paragraph of Related Work). **Insights**. Driven by the limitations identified above—**especially the neglect of deeper semantic and decision discrepancies between heterogeneous models**—this work introduces FedKWAZ, a cognitively-aware mutual learning framework that explicitly targets the problem of semantic–decision dual drift. It formalizes KWAZ as spatial zones where heterogeneous models display significant knowledge divergence, and further decomposes them into SWAZ and DWAZ, enabling targeted distillation in the semantic and decision discrepancy zones, respectively. In addition, a lightweight class-level global knowledge anchoring mechanism is designed to achieve a new balance among architectural flexibility, communication efficiency, and transfer quality.By directly confronting the **semantic–decision dual drift**, FedKWAZ bridges overlooked knowledge gaps and enables more precise and efficient cross-model collaboration.
>
> **Q1**: We appreciate the reviewer’s attention to the definition of $\epsilon$ in our convergence analysis and understand the concern regarding the potential ambiguity of “directly assuming the gradient norm is bounded by $\epsilon$.” To avoid misunderstanding, we would like to further clarify the logic of our analysis and the role of $\epsilon$.
>
> The derivation we adopt follows the **“target-driven”** analysis paradigm, which has been widely used in recent heterogeneous federated learning literature. The key idea is not to treat $\epsilon$ as a priori bound on the gradient norm, but rather to **first specify an acceptable convergence precision target $\epsilon$, and then derive the communication rounds $T$, learning rate $\eta$, and other hyperparameter conditions required to reach this target**. This approach has been adopted by several works on heterogeneous learning, such as FedProto [1].
>
> Specifically, we first establish the theoretical framework in Appendix Section C. Under a set of reasonable assumptions, Lemma 1 and Lemma 2 respectively provide the upper bounds for local training errors in the two stages: global semantic-decision alignment and KWAZ-guided mutual learning. Then, Theorem 1 combines them to derive the overall loss descent in a full communication round. Finally, in Theorem 2, we set an arbitrarily small precision target $\epsilon$, and derive the constraints on $\eta$ and $T$ such that the expected average squared gradient norm is below $\epsilon$, ensuring convergence to the desired accuracy.
>
> **It is important to emphasize that $\epsilon$ serves as a convergence target, not a prior assumption. It is introduced as a controllable error threshold to characterize the convergence behavior under a finite number of training rounds**.
>
> This target-driven analysis contrasts with the asymptotic approach, such as in FedAvg, which focuses on whether the gradient norm vanishes as $T \to \infty$. In contrast, our approach emphasizes whether there exists a feasible parameter configuration that allows the model to achieve a specified accuracy within limited communication cost, aligning better with the real-world deployment demands of federated learning.
>
> We fully agree with the reviewer on the importance of asymptotic analysis and plan to explore incorporating the two-stage non-convex optimization process of FedKWAZ into such a framework in future work. In the current revision, we will further clarify the role of $\epsilon$ and supplement our explanation by showing its consistency with the derivation paradigms used in works like FedProto.
>
> [1]FedProto: Federated Prototype Learning across Heterogeneous Clients. AAAI 2022

---

> > ### Comment · Reviewer_c2QV · 2025-08-04
> >
> > Thank you for the detailed response. I will stay with my positive rating.

---

> > > ### Author Response · Authors · 2025-08-05
> > >
> > > We sincerely appreciate your encouraging feedback and continued support. Thank you so much for your valuable comments!

---

### Official Review · Reviewer_pAYd · 2025-07-03

**Clarity:** 2
**Significance:** 3
**Originality:** 2
**Rating:** 4
**Confidence:** 3

**Summary:**

This paper proposes FedKWAZ, a cognitively-aware mutual learning framework for heterogeneous federated learning (HtFL). The authors identify a key limitation in existing federated mutual learning methods—namely, their reliance on direct output alignment between private and proxy models, which overlooks semantic and decision-level divergences. FedKWAZ introduces Knowledge Weak-Aware Zones (KWAZ), further divided into Semantic Weak-Aware Zones (SWAZ) and Decision Weak-Aware Zones (DWAZ), as the focal points of knowledge misalignment. The method consists of a dual-stage learning framework: (1) global class-level alignment via aggregated prototypes/logits, and (2) local KWAZ-guided distillation using Hierarchical Adaptive Patch Mixup (HAPM) and Knowledge Discrepancy Perceptron (KDP). Extensive experiments across multiple datasets and model heterogeneity levels demonstrate state-of-the-art results.

**Questions:**

See weakness

**Ethical Concerns:**

["NO or VERY MINOR ethics concerns only"]

**Final Justification:**

Thank for authors' rebuttal, which addressed my concerns. I will maintain my positive attitude and score.

**Limitations:**

The paper includes a dedicated limitations section (Appendix G), which acknowledges the potential cost in client-side computation and limited exploration of dynamic client behavior. These are reasonable and honestly addressed.

**Paper Formatting Concerns:**

Conforms to NeurIPS formatting. Figures and tables are legible.

**Quality:**

3

**Strengths And Weaknesses:**

**Strengths**:

- The paper addresses an under-explored but important challenge in federated mutual learning: misalignment between private and proxy models under heterogeneity.

- The concept of KWAZ is novel in the FL context and offers a structured way to locate and exploit knowledge discrepancies.

- The proposed HAPM and KDP modules are well-motivated and enhance the mutual learning process by focusing on semantic/decision-level disagreement zones.

- The experimental section is comprehensive, covering five datasets and three types of heterogeneity (data/model/client), with solid gains over strong baselines like FedTGP and FedKD.

- The paper includes theoretical convergence analysis, good ablation studies, and clear algorithmic structure.

**Weaknesses**:

- The conceptual novelty of KWAZ would be stronger if compared and contrasted more explicitly with existing saliency- or region-aware distillation methods in CV or KD literature.
- The paper lacks discussion and analysis of scalability to realistic FL scenarios involving thousands of clients or low-resource edge devices.
- The impact of key hyperparameters (patch granularity \(g\), mixing ratios \(\alpha, \beta\)) in HAPM is not analyzed or visualized.
- Some figures (especially Figure 1) are visually crowded and hard to parse; a cleaner design could improve clarity.
- Notations like \(\alpha^*, \beta^*, g^*\) are introduced abruptly and could be defined more clearly at first appearance.

---

> ### Author Rebuttal · Authors · 2025-07-30
>
> **W1**: We sincerely thank the reviewer for the valuable suggestion and agree that a more explicit comparison with existing saliency/region-aware distillation methods can further highlight the conceptual novelty of KWAZ.
>
> In knowledge distillation and computer vision tasks, the core idea of saliency or region-aware distillation [1,2,3] is to focus on the local regions of an image that contribute most to model prediction, thereby improving the efficiency of knowledge transfer. Such methods typically rely on attention maps or saliency responses generated inside the teacher model as guiding cues for student learning. The basic assumption is that the regions attended by the teacher represent the most discriminative semantic information, and aligning features or attention in those regions helps the student model acquire “effective knowledge” under limited capacity. However, these methods are essentially **“one-way guidance” — that is, the teacher statically determines which regions the student should learn from**.
>
> KWAZ fundamentally differs from the above methods in that it does not rely on the saliency judgment of a single model, but instead dynamically identifies the regions of cognitive discrepancy between the private model and the proxy model as alignment targets. This design has three innovations:
>
> First, traditional methods assume that the teacher's salient regions are optimal learning targets for the student, while KWAZ identifies regions with the largest representational or decision-level discrepancy between the two models. These regions often reflect knowledge that the student has not yet mastered. Distilling on these regions helps close knowledge gaps while **avoiding the neglect of student-specific errors, thereby making alignment more targeted**.
>
> Second, KWAZ automatically generates and evaluates mixed samples via HAPM+KDP to locate these discrepancy regions, without requiring saliency networks or hand-crafted rules. This is suitable for real-world federated heterogeneous mutual learning settings. While traditional saliency-based distillation relies on the **fixed authority of the teacher**, KWAZ operates in a **peer mutual-learning context** and focuses on bidirectional cognitive mismatch, making it more adaptable.
>
> Finally, KWAZ conceptually proposes a new perspective: **transforming inter-model cognitive discrepancies into learning opportunities**. Traditional methods focus on the “teacher's optimal regions,” whereas KWAZ focuses on the “most misaligned regions between student and teacher.” This shift in perspective enables more direct identification of performance bottlenecks and improves the quality of knowledge transfer, demonstrating both theoretical uniqueness and practical value.
>
> We hope the above comparison effectively addresses the reviewer’s concern regarding the novelty of KWAZ.
>
> **W2**: We sincerely thank the reviewer for the valuable suggestion and fully agree that discussing the scalability of FedKWAZ to ultra-large client populations and resource-constrained devices is essential for validating its practical applicability.
>
> Table 1: Test accuracy (%) on Tiny-ImageNet with 1000 clients under low client participation rates (ρ = 0.1).
>
> | Method    | $\rho=0.1$ |
> |----------|:----------:|
> | FedMRL   |   13.42    |
> | FedTGP   |   10.77    |
> | FedKTL   |   11.25    |
> | FedKWAZ  | **15.74** |
>
> To evaluate scalability, our main experiments included settings with 50, 100, and 200 clients, verifying the effectiveness of our method at moderate scales. In response to the reviewer’s concern regarding scenarios with thousands of clients, we conducted additional experiments on Tiny-ImageNet with 1000 clients and very low client participation rates (ρ = 0.1). As shown in **Table 1**, under this extremely sparse regime, FedKWAZ achieved 15.74% accuracy, significantly outperforming FedKTL (11.25%) and FedMRL (13.42%) at the low participation rates. These results suggest that **FedKWAZ can generate more transferable input cues via HAPM and effectively reinforce distillation in high-divergence regions**, thereby mitigating performance degradation even in large-scale, low-resource federated environments.
>
> Regarding low-resource devices, our main paper already demonstrates that FedKWAZ can **reduce both communication and computation costs by accelerating convergence**. In each round, only aggregated prototypes and logits are exchanged to align semantics and decisions, leading to low communication overhead—suitable for deployment on limited-resource devices. Furthermore, the HAPM and KDP modules are not triggered every round, but rather at fixed intervals (default $k=30$). When triggered, the client performs the search using only forward propagation over candidate configurations to identify those that maximize the divergence between models, without invoking backward propagation or weight updates, thus significantly reducing extra computation.
>
> On CIFAR-100, for instance, performing a KWAZ search once every 30 rounds leads to a per-round amortized overhead that accounts for only 8.93% of the local training computation, which is acceptable given the corresponding accuracy gain. Moreover, the trigger interval $k$ is tunable based on device capacity—larger $k$ values (i.e., less frequent searches) can further reduce computational cost, while still maintaining strong performance. For example, with $k=50$, FedKWAZ achieves 51.38% accuracy on CIFAR-100, outperforming the FedTGP baseline by 4.23%. Further optimization can also be achieved by reducing the size of the candidate search space.
>
> We thank the reviewer again for the detailed suggestion and will further refine the related discussion to strengthen the practical deployability of our method.
>
> **W3**: We thank the reviewer for their attention to the hyperparameter settings of HAPM. In our current design, $g$ controls the patch granularity (i.e., dividing an image into $\sqrt{g} \times \sqrt{g}$ patches), while $\alpha$ (chosen from $\\{0.1,\ 0.5,\ 1.0\\}$, following $\lambda^{(i,j)} \sim \text{Beta}(\alpha, \alpha)$) determines the mixing ratio between two images within each region. A larger $g$ leads to finer perturbations and stronger diversity, whereas a smaller $g$ helps preserve global semantics. A smaller $\alpha$ emphasizes inter-image differences, facilitating the induction of cognitive discrepancies, while a moderate value better preserves semantic consistency and training stability.
>
> **Our current configuration balances perturbation strength, semantic fidelity, and alignment efficiency**. As detailed in Appendix D.3, we set dataset-specific values of $g$ and $\alpha$ based on dataset characteristics. In response to the reviewer’s suggestion, **we will add visualizations of samples generated under different $g$ and $\alpha$ values in the revised version**, clearly illustrating how these hyperparameters modulate HAPM’s behavior and supporting the rationale of our parameter choices.
>
> **W4**: We sincerely thank the reviewer for the valuable suggestions regarding the figure design. We fully agree that the current Figure 1 contains a high density of information, which may hinder intuitive understanding. The reviewer’s suggestions are highly insightful for improving the clarity of the illustration. In the revised version, we will enhance the **color scheme** (to improve contrast between modules), adjust the **layout** (by simplifying arrow paths and emphasizing structural hierarchy), and enlarge the **label fonts** (to improve readability), aiming to present a more intuitive and concise visual representation. We believe these adjustments will substantially improve the reader’s experience and more clearly convey the problem setting and the core mechanism of FedKWAZ.
>
> **W5**: We thank the reviewer for pointing out the unclear definition of the symbols. We noticed that $(\alpha, g)$ first appears in the Introduction, and we have revised the corresponding description as follows:
>
> **“HAPM generates multiple mixed samples via diverse parameter pairs $(\alpha_i, g_i)$ (where $i=1, ..., K$), where $\alpha$ controls the mixing ratio between two images and $g$ specifies patch granularity to divide each image into $\sqrt{g} \times \sqrt{g}$ blocks for local mixing. KDP then evaluates representation and decision discrepancies between private and proxy models across these samples and selects $(\alpha\^\*, g\^\*)$ inducing the maximal divergence.”**
>
> This revision clarifies: (1) that $(\alpha_i, g_i)$ denotes a set of candidate mixing parameters; (2) the functional roles of each parameter; and (3) that $(\alpha\^\*, g\^*)$ refers to the optimal pair automatically selected by KDP as the target configuration for subsequent distillation. We thank the reviewer again for highlighting this important detail.
>
> [1] Paying More Attention to Attention: Improving the Performance of Convolutional Neural Networks via Attention Transfer. ICLR 2017
>
> [2] Robust Saliency-Aware Distillation for Few-shot Fine-grained Visual Recognition. IEEE Transactions on Multimedia 2024
>
> [3] Region-aware knowledge distillation between monocular camera-based 3D object detectors. ICT Express 2025

---

> > ### Comment · Reviewer_pAYd · 2025-08-06
> > **Official comments by Reviewer pAYd**
> >
> > Thank you to the authors for the rebuttal, which addressed my concerns. Therefore, I will keep my score unchanged

---

> > > ### Author Response · Authors · 2025-08-08
> > >
> > > We truly value your kind feedback and ongoing support. Thank you greatly for your insightful comments.

---

### Official Review · Reviewer_nXns · 2025-07-03

**Clarity:** 3
**Significance:** 3
**Originality:** 3
**Rating:** 4
**Confidence:** 3

**Summary:**

The paper proposes FedKWAZ, a novel federated learning framework designed to address dual heterogeneity in both non-IID data and diverse client models through key innovations including: Knowledge Weak-Aware Zones (KWAZ) that identify regions of maximal discrepancy between client-private and proxy models by decomposing them into Semantic Weak-Aware Zones (SWAZ) for feature-space misalignment, and Decision Weak-Aware Zones (DWAZ) for output-space misalignment. Hierarchical Adaptive Patch Mixing (HAPM) that generates mixed samples to probe model discrepancies. A Knowledge Discrepancy Perceptron (KDP) that selects samples with largest SWAZ/DWAZ gaps for targeted distillation, and a two-stage training approach where Stage I establishes global class-level alignment using aggregated prototypes while Stage II implements local KWAZ-guided bidirectional distillation. Experimental validation across five datasets (CIFAR-10/100, Flowers102, etc.) demonstrates that FedKWAZ outperforms ten baselines by 2.45–4.66% in accuracy under high heterogeneity conditions, scales effectively to 200 clients, and reduces communication costs by 3–10× compared to proxy-sharing methods.

**Questions:**

See weaknesses for details.

**Ethical Concerns:**

["NO or VERY MINOR ethics concerns only"]

**Limitations:**

None.

**Paper Formatting Concerns:**

None.

**Quality:**

3

**Strengths And Weaknesses:**

Strengths
1. KWAZ provides a granular framework to pinpoint and rectify cross-model cognitive gaps, advancing beyond crude output alignment.
2. Technical Innovation: HAPM and KDP enable dynamic discovery of hard-to-align samples, while the two-stage design decouples global consistency from local refinement.
3. Communication overhead is slashed by avoiding proxy model transmission, using lightweight prototypes instead.
4. Convergence proofs (Appendix C) guarantee non-convex optimization at rate O(1/T).
5. Code/dataset links, hyperparameters, and ablation studies (e.g., SWAZ/DWAZ impact) are thoroughly documented.

Weaknesses
1. Temperature (τ=4) and feature dimension (K=512) are fixed. Exploring adaptive schemes (e.g., τ tuned per model complexity) could boost performance.
2. Experiments use CNNs/ViTs but lack extreme architectural gaps (e.g., transformers vs. MLPs). Testing cross-backbone distillation (e.g., CNN→ViT) would strengthen claims.
3. While upload costs are reduced, the overhead of frequent HAPM/KDP operations (every k=30 rounds) is not quantified.
4. Privacy/fairness risks of shared prototypes (e.g., membership inference) are noted in Appendix H but lack mitigation strategies.

---

> ### Author Rebuttal · Authors · 2025-07-30
>
> **W1**: We sincerely thank the reviewer for the insightful suggestion. The idea of dynamically adjusting the distillation temperature ($\tau$) and the feature alignment dimension ($K$) based on model complexity or data characteristics offers a highly promising direction for performance enhancement.In our current experiments, we adopt a fixed $\tau = 4$ and $K = 512$ to ensure consistent experimental settings. However, the reviewer’s observation has made us more aware of the potential benefits of adaptive strategies, especially under heterogeneous settings. Recent studies in knowledge distillation, such as [1], [2], and [3], have explored adaptive temperature scheduling and logit scaling, further validating the feasibility and value of the reviewer’s suggestion.
>
> Table 1: Accuracy on CIFAR-100 in Practical Settings with Different Values of $\tau$.
>
> |               | $\tau=2$     | $\tau=3$      | $\tau=4$      | $\tau=5$        | $\tau=6$
> |---------------|------------|------------|------------|------------|------------|
> | Accuracy(%)     | 50.23 | 50.97 | 51.69 | 51.44 | 51.38 |
>
> To further investigate, we conducted additional experiments evaluating the effect of different $\tau$ values in FedKWAZ. As shown in **Table 1**, on the CIFAR-100 dataset, FedKWAZ achieves accuracy of 50.23%, 50.97%, 51.44%, and 51.38% for $\tau = 2, 3, 5, 6$, respectively, confirming $\tau = 4$ as the optimal setting. This trend suggests that small $\tau$ may produce overly sharp distillation signals, while large $\tau$ may introduce noise, thus weakening the transfer effectiveness. These results highlight the importance of properly tuned temperature in heterogeneous knowledge transfer.
>
> As a next step, we plan to incorporate learnable or schedule-driven temperature adjustment mechanisms, and explore strategies to dynamically set the feature alignment dimension $K$ based on model capacity or data distribution. This direction will serve as a key extension of our framework. We truly appreciate the reviewer’s constructive feedback.
>
> **W2**: We thank the reviewer for the valuable suggestion and have strengthened our claim by demonstrating the effectiveness of our method under scenarios involving greater architectural heterogeneity. In the $HtM_{10}$ heterogeneous setup, we explore a cross-backbone distillation setting from CNN to ViT. Specifically, in $HtM_{10}$, each client maintains a 4-layer CNN as its proxy model for knowledge exchange, while some clients adopt ViT architectures (including ViT-B/16 and ViT-B/32) as their private models. This setup enables bidirectional knowledge transfer across different backbones, such as CNN-to-ViT and ViT-to-CNN, where our method performs notably well.
>
> To further assess the robustness of our method under even more extreme architectural disparities, we conducted an additional experiment **where each client uses a multi-layer perceptron (MLP) as its proxy model**. The MLP consists of three hidden layers with dimensions 1000, 500, and 200, followed by a projection layer mapping to a unified 512-dimensional feature space, and a final fully connected classification layer—representing a standard MLP structure. The private client models span a wide range of architectures, including 4-layer CNNs, GoogleNet, MobileNetV2, ResNet variants (18/34/50/101/152), and ViT-B/16 & ViT-B/32.
>
> Table 2: Test accuracy (%) on Cifar100 and Flowers102 datasets using MLP as the proxy model.
>
> | Method    | Cifar100 | Flowers102 |
> |----------|:--------:|:----------:|
> | FedMRL   |  41.98   |   48.22    |
> | FedTGP   |  47.15   |   53.72    |
> | FedKTL   |  47.09   |   52.85    |
> | FedKWAZ  | **51.24** | **57.31** |
>
> This distillation setting, involving Transformer-to-MLP transfer, presents a significant challenge for cross-backbone knowledge alignment. Experimental results on both CIFAR-100 and Flowers102 datasets demonstrate the strong performance of FedKWAZ, which achieves 51.24% and 57.31% accuracy, respectively. Compared to baseline methods such as FedMRL (41.98%, 48.22%) and FedKTL (47.09%, 52.85%), FedKWAZ delivers substantial improvements. These results validate the robustness of our approach in handling architectural heterogeneity between teacher and student models. We sincerely thank the reviewer for the constructive suggestion, which has enabled us to further strengthen our experimental design.
>
> **W3**: We thank the reviewer for the opportunity to further clarify the computational cost of HAPM and KDP. These modules are not executed in every round but are instead triggered periodically (e.g., every $k = 30$ rounds by default). When a HAPM/KDP cycle is activated, each client performs a lightweight search over a predefined set of candidate parameter configurations to identify the one that maximizes the knowledge discrepancy between the local and proxy models. Crucially, this search involves only forward propagation, without any backward propagation or weight updates, and thus incurs limited additional overhead.
>
> Based on our measurements on CIFAR-100, inserting a KWAZ  search step every 30 rounds introduces an extra cost of $4.75 \times 10^{4}$ GFlops, which averages to $1.58 \times 10^{3}$ GFlops per round—only **8.93%** of the total per-round computation. This moderate overhead is a reasonable trade-off given the significant accuracy gains it enables.
>
> Moreover, $k$ (i.e., the search frequency) is a tunable hyperparameter, allowing users to reduce the overhead by increasing $k$ (i.e., triggering HAPM/KDP less frequently), with only marginal impact on performance. In our hyperparameter study, FedKWAZ maintained strong performance even at $k=40$ or $k=50$. For example, at $k=50$, FedKWAZ achieved 51.38% accuracy on CIFAR-100, outperforming the FedTGP baseline by 4.23%. Increasing $k$ to 40 and 50 reduces the amortized per-round KDP-related computation by 25% and 40%, respectively, compared to the default setting of $k=30$. This shows that users can flexibly adjust $k$ according to device capabilities, while still enjoying efficient and effective knowledge transfer.
>
> We appreciate the reviewer’s insightful question, which allowed us to better articulate the practicality of our method.
>
> **W4**:We sincerely thank the reviewer for highlighting the importance of ensuring data privacy and fairness in federated learning. Our method has been carefully designed with several measures to mitigate privacy risks. Specifically, we only transmit aggregated class-level prototypes and prediction distributions, rather than raw data or complete model weights. This strategy avoids directly exposing client training samples or model parameters and aligns with a number of privacy-conscious FL methods such as [4,5], which also opt to share intermediate statistics rather than full model updates.
>
> Regarding fairness, we deliberately avoid relying on client-reported local dataset sizes for weighted aggregation. Traditional methods like FedAvg often require the number of samples per client to compute weighted averages, which may inadvertently reveal data-related information and grant undue influence to data-rich clients, potentially leading to global knowledge bias. In contrast, our aggregation of prototypes and logits treats each client’s contribution to a class $c$ as equal, without requiring the server to access any information about client data volume:
>
> $$
> \\bar{P}\_c\^{\\mathcal{M}} = \\frac{1}{|\\mathcal{N}\_c|} \\sum\_{k \\in \\mathcal{N}\_c} P\_c\^{\\mathcal{M}\_k},
> \\bar{L}\_c\^{\\mathcal{M}} = \\frac{1}{|\\mathcal{N}\_c|} \\sum\_{k \\in \mathcal{N}\_c} L\_c\^{\\mathcal{M}\_k}
> $$
>
>
> This design both protects the privacy of data quantity and distribution and avoids the systemic bias introduced by unbalanced data. Furthermore, by not transmitting full model structures or parameters, our approach is inherently more resilient to certain types of privacy attacks compared to conventional FL frameworks.
>
> We also acknowledge the reviewer’s valid concern that even aggregated class-level prototypes or logits might be subject to potential privacy risks such as membership inference. To address this, we plan to incorporate differential privacy mechanisms in future work (e.g., injecting calibrated noise into the shared prototypes or logits), or explore other perturbation-based strategies to further reduce the risk of information leakage while maintaining distillation effectiveness. We will expand the discussion in the final version of the paper and clearly articulate these planned enhancements.
>
> Once again, we thank the reviewer for bringing attention to these critical concerns. We remain committed to improving the design of our framework to proactively tackle privacy and fairness challenges in federated settings.
>
> [1] Curriculum Temperature for Knowledge Distillation. AAAI 2023
>
> [2] Logit Standardization in Knowledge Distillation. CVPR 2024
>
> [3] Rethinking the Temperature for Federated Heterogeneous Distillation. ICML 2025
>
> [4] Federated Learning from Pre-Trained Models: A Contrastive Learning Approach. NeurIPS 2022
>
> [5] FedTGP: Trainable Global Prototypes with Adaptive-Margin-Enhanced Contrastive Learning for Data and Model Heterogeneity in Federated Learning. AAAI 2024

---

### Note · Authors · 2025-08-12

We sincerely thank you for the valuable time and professional feedback in reviewing our paper. Your suggestions **(covering method effectiveness verification, practicality analysis, and clarity)** have greatly refined our work. Below is a concise summary:

## **Contributions of our Paper**

This work targets HtFL with dual heterogeneity in data and models, revealing the **semantic–decision dual drift** between private and proxy models in federated mutual learning and its impact on knowledge transfer.

To address this, we propose FedKWAZ, which **identifies the largest knowledge-gap zones via KWAZ and bridges semantic and decision gaps through SWAZ and DWAZ**. It adopts a two-stage strategy: Stage I **coarsely aligns global semantic prototypes and decision anchors**; Stage II leverages **HAPM and KDP for structure-aware bidirectional distillation in KWAZ**.

Across **two data- and six model-heterogeneity settings with 14 architectures (e.g., ResNets and ViTs)**, FedKWAZ outperforms 10 advanced methods on multiple benchmarks.

## **Summary of our Rebuttal**

**HAPM effectiveness and dynamics**: Compared with top 10%, 30%, and 50% hard-sample-enhanced schemes, HAPM better amplifies knowledge differences and strengthens mutual learning **(+4.08% on CIFAR-10, +10.85% on Flowers102)**, surpassing all hard-sample-based methods. Its dynamic parameters ($\alpha, \beta, g$) vary widely between clients yet remain stable within clients, showing adaptability to structural heterogeneity.

**Robustness and scalability**: In MLP proxies, 1,000 clients, low participation, and high local rounds, FedKWAZ remains stable, confirming applicability in highly complex environments.

**Computation overhead**: With k=30, HAPM/KDP uses only **8.93%** of local computation per round; increasing k cuts costs by 25–40% while preserving performance. Compared with FML and FedTGP, FedKWAZ gains **4.23–11.91%** accuracy with low cost.

**Design motivation and differentiation**: Clarifies the two-stage mutual learning logic and, from a semantic–decision alignment perspective, positions FedKWAZ among related works; clearly distinguishes it from saliency/region-aware distillation in target localization.

**After discussion, reviewers unanimously gave positive evaluations and recommended including the discussed analyses and improvements in the final version. We will follow this and present all relevant experiments and optimized expressions in the final manuscript**.

Best regards,

The Authors

---

### Decision · Program_Chairs · 2025-09-17

**Decision:**

Accept (poster)

**Comment:**

This paper proposes a novel framework for heterogeneous federated learning. It integrates four key mechanisms into a two-stage mutual learning framework to address both data and architectural heterogeneity.  This paper received four borderline acceptances, and most concerns about experiment details were adequately addressed during the rebuttal. Therefore, it is recommended for acceptance.